# Generalised Hodgkin–Huxley model captures human P2X and AMPA receptor currents

Alireza Poshtkohi[1] [iD] and Brian D. Gulbransen[2] [iD]

[1] *School of Physics, Engineering and Computer Science, University of Hertfordshire, Hatfield, Hertfordshire, UK*
[2] *Department of Physiology, Michigan State University, East Lansing, Michigan, USA*

Handling Editors: Kim Barrett & Brian Delisle

The peer review history is available in the Supporting Information section of this article (https://doi.org/10.1113/JP288880#support-information-section).

**Abstract figure legend** A universal, biophysically interpretable mathematical model for human ionotropic receptors. The generalised Hodgkin–Huxley (gHH) model unifies the mathematical description of all major ligand-gated

**Alireza Poshtkohi** is a computational neuroscientist and applied mathematician who develops innovative mathematical and computational frameworks to address complex problems at the intersection of biology, medicine, engineering and physics by using the principles of data-driven model discovery. He holds a Ph.D. in computational neuroscience from Ulster University and a B.Sc./M.Sc. in electrical and electronics engineering, and currently lectures in computer science at the University of Hertfordshire. Dr Poshtkohi's research spans high-performance computing (HPC), mathematical modelling, biophysics and systems physiology, with particular focus on unifying biophysical principles to understand the human brain, neuroimmune and enteric nervous systems. His work includes developing large-scale simulation environments for neuroscience, constructing generalised biophysical models of ion channels and building collaborative computational-experimental platforms for drug discovery and disease modelling. He is the author of a textbook on high-performance computing; has published in leading journals across computational biology, neuroscience, and computer engineering; and actively collaborates with experimental physiologists in the UK and US.

ionotropic receptors (P2X$_{1-7}$, AMPA) across diverse cell types, capturing activation, inactivation, desensitisation and recovery dynamics with a single, interpretable structure. The gHH framework links experimental recordings to model parameters, providing a universal, biophysically interpretable and predictive framework with applications in neuroscience, drug discovery and neurophysiological modelling, while overcoming the complexity and non-identifiability issues of multi-state Markov models.

**Abstract** Ionotropic receptors are transmembrane ion channels that play central roles in regulating synaptic transmission in the nervous system and cellular activity underlying immune responses. However, a unified mathematical model capturing their dynamics remains elusive. In this paper a generalised Hodgkin–Huxley (gHH) model is introduced, which seamlessly represents different activation, inactivation and recovery dynamics of the entire human P2X receptor family and the human AMPA-type glutamate receptor. The model incorporates two activation gates ($m_1$, $m_2$) and two inactivation gates ($h_1$, $h_2$) to connect electrophysiological recordings to the underlying kinetics of ligand-gated receptor currents beyond voltage-gated channels. We propose five distinct forms of whole-cell currents to describe the gating kinetics of ion channels. The model takes receptor-specific cooperativity, binding kinetics and desensitisation pathways into account. Validation using a wide range of datasets demonstrates the model's robustness in quantitatively predicting receptor responses. It is shown that the framework exhibits multi-scale temporal dynamics by which rapid activation and prolonged recovery are seamlessly captured, ranging from milliseconds and seconds to minutes. Notably, the model replicates the prolonged ATP-dependent recovery time of hP2X$_3$ receptor over several minutes and the millisecond recovery time of hGluA1 receptor reported experimentally. This work provides a single mathematical structure by parametrising the kinetics of all major human ionotropic receptors, thereby providing a universal, biophysically interpretable and predictive framework with applications in neuroscience, drug discovery and neurophysiological modelling. It also represents a closer step towards a unified theory of electrophysiological modelling for understanding ion channel function in health and disease.

(Received 13 March 2025; accepted after revision 3 October 2025; first published online 4 November 2025)

**Corresponding author** A. Poshtkohi: School of Physics, Engineering and Computer Science, University of Hertfordshire, Hatfield, Hertfordshire, UK. Email: a.poshtkohi@herts.ac.uk

**Key points**

- The Hodgkin–Huxley model was generalised to ligand-gated receptors beyond voltage-gated dynamics.
- The framework offers a unified, biophysically interpretable mathematical structure to capture the gating properties of human ionotropic receptors (validated against experimental data from hP2X$_{1-7}$ and hGluA1 receptors).
- The model provides quantitative insights into how P2XRs and AMPA receptors control ion channel function.
- The proposed tool establishes an in-silico modelling infrastructure with applications in synaptic physiology, neuroinflammation and drug discovery.

# Introduction

The Hodgkin–Huxley (HH) model originally published in *The Journal of Physiology* 73 years ago remains a fundamental mathematical framework in electrophysiology (Catterall et al., 2012; Hodgkin & Huxley, 1952). It has shaped modern neuroscience, computational biology and neuropharmacology. The original HH model describes voltage-gated ion channels through two independent activation and inactivation gates and calculates the current using a power law dependency. However the HH formalism assumes independent gating processes, which makes it inadequate to capture the dynamics of ligand-gated receptors. Agonist binding, cooperative subunit interactions and state-dependent inactivation are inherently coupled in ligand-gated receptors (Dixon et al., 2022).

Almost shortly after Hodgkin and Huxley published their work, Markov models (MMs) were proposed (Armstrong, 1971) in an attempt to tackle these limitations. Since then they have been extensively used to model ligand-gated receptors such as rodent P2X purinergic receptors (8–64 states, dozens of parameters) (Keceli & Kubo, 2014; Khadra et al., 2012, 2013; Mackay et al., 2017; Yan et al., 2010; Zemkova et al., 2015), rodent AMPA/NMDA glutamate receptors (Destexhe et al., 1998; Smith et al., 2000), rodent multi-ligand-gated receptors (De Young & Keizer, 1992; Sneyd & Dufour, 2002) and human voltage-gated ion channels (Asfaw & Bondarenko, 2019; Balbi et al., 2017; Trayanova et al., 2023). Despite their detailed receptor kinetics, MMs result in multiple challenges such as high-dimensional parameter spaces, complex state topologies, assumptions on state transitions and computational intractability. They make MMs impractical for whole-cell simulations and introduce intricacies for parameter estimation (Mangold et al., 2021; Siekmann et al., 2011; Linaro & Giugliano, 2022). Constructing an optimal Markov structure that can accurately capture the behaviours of the underlying ion channels is difficult. This is more critical for ion channels in which multiple activation/inactivation and desensitisation pathways exist. MM models can also limit interpretability and predictability because they contain many parameters and state variables that lead to parameter unidentifiability issues (Siekmann et al., 2011).

Herein a generalised Hodgkin–Huxley (gHH) is introduced to address these challenges by directly extending HH formalism to human ligand-gated receptors. The model supports complex biophysical mechanisms which include interdependencies between activation and inactivation, agonist concentration on model parameters and cooperative gating (Ding & Sachs, 2002; Dixon et al., 2022; Michel et al., 2007; Wang & Yu, 2016). The model is also interpretable and scalable. Therefore it preserves two key features of the original HH model unlike Markov models: simplicity and biophysical transparency. However, in contrast to the HH model, the gHH model captures different intricate gating properties of human receptors. Specifically the model extends the HH model in three fundamental directions: (1) two activation gates ($m_1$, $m_2$) and two inactivation gates ($h_1$, $h_2$) enable a quantitative description of subunit-specific contributions to channel opening and desensitisation, (2) power-law exponents ($n_1 - n_7$) make it feasible to capture cooperative ligand binding and agonist-dependent modulation of gating kinetics in a mathematically robust manner and (3) five distinct current formulations (multiplicative, additive, hybrid) allow the model to be adapted to receptor-specific stoichiometries and kinetics for ensuring universal applicability.

The gHH model retains the mathematical elegance in HH formalism while addressing its historical constraints. Ionotropic receptors play critical roles in neurotransmission, immune responses and glial and cardiovascular physiology (Ceprian & Fulton, 2019; Gallina et al., 2021; Illes et al., 2020; North, 2002; Ralevic, 2015; Seguella & Gulbransen, 2021; Selezneva et al., 2022). Changes to ion channel function are directly linked to pathological conditions (Dixon et al., 2022). Therefore an accurate and scalable framework for human receptor modelling is essential for understanding normal and pathological conditions. The model architecture described herein accurately captures the dynamics of ligand-gated human receptors, including ATP-gated purinergic P2X receptors and glutamate-gated AMPA receptors. It will be shown that the model can accurately reproduce biphasic currents, desensitisation plateaus and recovery kinetics observed in human experimental data. Unlike MM-based approaches it avoids the state explosion problem, making it computationally feasible for whole-cell and network-scale simulations in systems neuroscience and drug discovery. The model is validated by fitting its parameters and choosing the best current equation against multiple human datasets for the entire P2X receptor family (hP2X$_{1-7}$) and the AMPA receptor (hGluA1).

Our work represents a closer step towards a unified theory of ion channel modelling in electrophysiology through a scalable, interpretable and computationally efficient approach. The gHH model can capture multi-scale gating properties that range from millisecond activation to minute-scale receptor desensitisation. These features demonstrate the model's potential as a universal framework in modelling ion channels and receptors with broad implications for biophysics, translational medicine and pharmacological targeting of receptors.

## Methods

In this section, the gHH framework is developed that models human ionotropic receptors with a strong focus on P2X (hP2X$_{1-7}$) and AMPA (hGluA1) variants. Figure 1*A* illustrates the expression of these receptors on neurons (central and peripheral), microglia, astrocytes, macrophages and enteric glia. These receptors play significant roles in ATP-gated and glutamate-mediated signalling in the brain and brain-like neurocircuitry of the enteric nervous system in the gut.

The model incorporates two activation gates ($m_1$, $m_2$) and two inactivation gates ($h_1$, $h_2$) that capture receptor gating dynamics in which rate constants are functions of agonist concentrations either ATP or Glu. The architecture shown in Fig. 1*B* provides a high-level view of the model in terms of interactions between gating

variables and the induced receptor current ($I_{hIR}$). The model allows for precisely replicating a wide spectrum of receptor dynamics consisting of activation, inactivation, desensitisation and recovery in different physiological conditions.

## The gHH model equations

We denote by $m_1$ and $m_2$ two *activation* (or 'open') gating variables and by $h_1$ and $h_2$ two *inactivation* (or 'closing') gating variables. Each of these variables, $m_i$ or $h_i$, is dimensionless and varies between 0 and 1. A value of 0 indicates that none of the channels are in the corresponding activated/inactivated state; a value of 1 indicates that all channels are in that state. The model dynamics is written as four simultaneous first-order equations:

$$\frac{dm_1}{dt} = \alpha_{m_1}(A) \times A \times (1 - m_1) - \beta_{m_1}(A)$$
$$\times [1 + \varphi(A)] \times m_1 \tag{1}$$

$$\frac{dm_2}{dt} = \alpha_{m_2}(A) \times m_1^{n_1} \times (1 - m_2) - \beta_{m_2}(A)$$
$$\times [1 + \varphi(A)] \times m_2 \tag{2}$$

$$\frac{dh_1}{dt} = \alpha_{h_1}(A) \times [1 + \varphi(A)] \times (1 - h_1) - \beta_{h_1}(A)$$
$$\times h_1 \times m_1^{n_2} \tag{3}$$

$$\frac{dh_2}{dt} = \alpha_{h_2}(A) \times [1 + \varphi(A)] \times (1 - h_2) - \beta_{h_2}(A)$$
$$\times h_2 \times m_2 \times m_1^{n_3} \tag{4}$$

These equations directly extend the HH framework to capture the kinetics of a variety of human ionotropic (e.g. purinergic or glutamatergic) receptors. $A$ is the agonist concentration (e.g. ATP or glutamate), $\varphi(A)$ is an additional modulation factor (e.g. $\varphi(A) = e^{-A}$ for AMPA receptors, and 0 for P2X receptors) and $n_1$, $n_2$, $n_3$ are integer exponents that differ by receptor subtype. Note that the units for agonist ($A$), rate constants

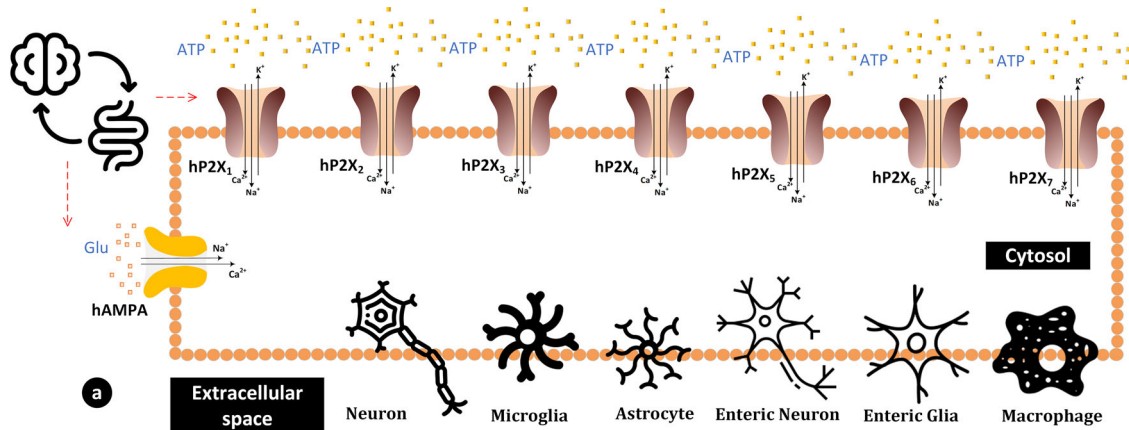

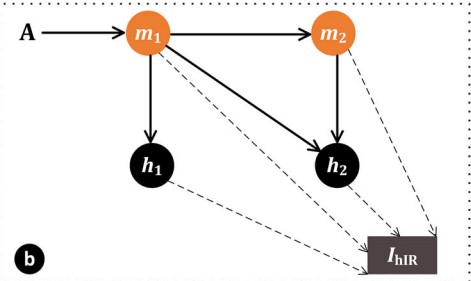

**Figure 1. Visual representation of human P2X and GluA1 receptors and model dynamics**
*A*, distribution of human P2X receptor subtypes (hP2X$_1$, hP2X$_2$, hP2X$_3$, hP2X$_4$, hP2X$_5$, hP2X$_6$ and hP2X$_7$) and human AMPA (GluA1) on various cell types. *B*, architecture of the gHH model adapted for ionotropic receptors by depicting the complex interactions of state variables which are influenced by the agonist *A* (either ATP or Glu). $m_1$ and $m_2$ denote activation gates, whereas $h_1$ and $h_2$ stand for inactivation gates. The induced current is denoted by $I_{hIR}$ and reflects the dynamic outcomes of receptor activation and modulation by the agonist. Note that the dotted lines show which gates the current depends on. The model structure can flexibly replicate intricate interactions experimentally observed in gating kinetics in response to varying levels of agonist concentration. It is also indicative of the diverse functionalities of ionotropic receptors.

and time are given separately for each receptor model in the results section and Supporting Information S1. We selected $\varphi(A) = e^{-A}$ because it is the simplest one-parameter, monotonic function that (i) equals 1 at agonist wash-out ($A \to 0$) and therefore boosts the recovery/de-inactivation ($\beta_{m_i}$ and $\alpha_{h_i}$) rates by a factor of two, and (ii) decays smoothly to 0 at high agonist, so the backward terms revert to their baseline forms. This dual behaviour lets the four-ODE gHH system reproduce the fast recovery of AMPA receptors after brief pulses without adding extra states or rate constants.

Each rate constant $\alpha_{m_i}(A), \beta_{m_i}(A), \alpha_{h_i}(A), \beta_{h_i}(A)$ is typically fitted to experimental data currents. Biologically, these rate constants control how rapidly channels transition between open/closed or active/inactive states in response to agonist binding.

The forward/activation term $\alpha_{m_1}(A) \times A \times (1 - m_1)$ in (1) models the opening of the activation gate $m_1$. When the agonist $A$ is present, it drives $m_1$ upward from 0 towards 1. The term $(1 - m_1)$ ensures that if $m_1$ is already close to 1, the remaining current for further activation is reduced. The desensitisation (or deactivation) term $\beta_{m_1}(A) \times [1 + \varphi(A)]$ in eqn (1) pulls $m_1$ back downward. Once gate $m_1$ is in an activated state, it can desensitise or deactivate with rate $\beta_{m_1}(A)$. The term $[1 + \varphi(A)]$ captures the extra inactivation or desensitisation effect that can occur for certain receptors (e.g. AMPA) if $\varphi(A) \neq 0$. $m_2$ has a structure close to $m_1$ in eqn (2) but depends on $m_1^{n_1}$ to capture cooperativity or sequential gating whereby the activation of $m_2$ requires $m_1$ to be partially activated. The desensitisation term again includes the $\beta_{m_2}(A) \times [1 + \varphi(A)] \times m_2$ factor.

The function of inactivation gates is more complex. When $m_1$ and/or $m_2$ are *small* (i.e. the channel is not strongly activated), the 'de-inactivation' term $\alpha_{h_i}(A) \times [1 + \varphi(A)] \times (1 - h_i)$ dominates. Because $h_i$ is close to 1, $(1 - h_i)$ is small – but as soon as $h_i$ drops even slightly below 1, this first term acts to pull $h_i$ back up toward 1. In other words, if activation (through $m_1$ or $m_2$) is low, $h_i$ relaxes to its baseline level of 1, signifying *no inactivation*. The second term

$$\beta_{h_i}(A) \times h_i \times \begin{cases} m_1^{n_2} & \text{(eqn (3))} \\ m_2 \times m_1^{n_3} & \text{(eqn (4))} \end{cases}$$

*reduces* $h_i$ away from 1 (i.e. it drives $h_i$ downward towards partial or full inactivation). In eqn (3), the factor $m_1^{n_2}$ means that if $m_1$ becomes large (strong channel activation), inactivation ramps up quickly and $h_1$ drops below 1. The product $m_2 \times m_1^{n_3}$ in eqn (4) similarly couples inactivation to *both* $m_2$ and $m_1$. If these activation gates rise, $h_2$ is driven downward from 1.

If $m_1$ and $m_2$ *fall back to zero* – for instance, when the agonist is removed – then the inactivation term shrinks. This implies that the first de-inactivation term $\alpha_{h_i}(A) \times [1 + \varphi(A)] \times (1 - h_i)$ becomes dominant and pushes $h_i$

back up towards 1. This biophysical mechanism signifies the recovery (also known as de-inactivation) process by allowing the channel to return to a non-inactivated state once strong activation is gone.

As before $\alpha_{h_i}(A)$ and $\beta_{h_i}(A)$ are kinetic rate constants that can depend on the agonist level. $\varphi(A)$ (e.g. $\varphi(A) = e^{-A}$ for certain receptors) can *boost* or *modify* the de-inactivation rate when the agonist is present. This modulatory factor is assumed to be equal to zero for those receptors whose extra fast desensitisation is not necessary in order to minimally reflect the gating dynamics.

In the traditional HH model currents flowing through a population of channels can be represented by:

$$I_{ion} = g_{max} \times (\text{'gating expression'}) \times (V_m - V_r) \quad (5)$$

where $V_m$ is the membrane potential, $V_r$ is the reversal (or Nernst) potential, and the *gating expression* typically involves products (and powers) of activation/inactivation variables that range between 0 (fully closed or inactivated) and 1 (fully open or inactivated).

In the gHH model, this idea is generalised by allowing *either* sums *or* products (*or* their combinations) over the gating variables $\{m_1, m_2, h_1, h_2\}$ which differ based on the receptor subtype under study. This flexible formulation of ionic currents makes capturing the distinct experimentally observed kinetic properties of a wide family of human ionotropic receptors practical.

The first formulation of eqn (5) using this generalisation is given by eqn (6):

$$I_{hIR_1} = g_{hIR} \times (m_1 + m_2)^{n_4} \times h_1^{n_5} \times h_2^{n_6}$$
$$\times (V_m - V_r) \quad (6)$$

The sum $(m_1 + m_2)$ in the activation term $(m_1 + m_2)^{n_4}$ implies that channels can open if *either* $m_1$ or $m_2$ (or both) is activated. The exponent $n_4$ captures the cooperativity or 'steepness' of this summed effect. Both $h_1$ and $h_2$ enter multiplicatively in the inactivation term $h_1^{n_5} \times h_2^{n_6}$, meaning that *each* inactivation gate can reduce the current. This combination – *additive activation* and *multiplicative inactivation* – can represent receptors that have parallel, independent activation subunits and serial-like inactivation gates.

The second form of eqn (5) is proposed by eqn (7):

$$I_{hIR_2} = g_{hIR} \times m_1^{n_4} \times m_2^{n_5} \times (h_1 + h_2)^{n_6}$$
$$\times (V_m - V_r) \quad (7)$$

Here, *both* $m_1$ and $m_2$ in the activation term $m_1^{n_4} \times m_2^{n_5}$ contribute multiplicatively, implying a more *cooperative* opening: the channel passes significant current only when *both* activation gates are high. The exponents $(n_4, n_5)$ again tune the level of cooperativity. The sum $(h_1 + h_2)$ implies that if *either* inactivation gate is engaged, it can significantly reduce/desensitise current. This summed

expression can mimic partial or complete inactivation of specific receptors, depending on the exponents used, when either gate transitions to an inactive state. This form – *multiplicative activation* and *additive inactivation* – is also useful for channels that require simultaneous activation steps but have multiple parallel inactivation mechanisms.

Equation (8) presents the third form of eqn (5):

$$I_{hIR_3} = g_{hIR} \times (m_1 + m_2)^{n_4} \times (h_1 + h_2)^{n_5}$$
$$\times (V_m - V_r) \tag{8}$$

As in $I_{hIR_1}$ *either* $m_1$ or $m_2$ can open the channel, with an exponent controlling cooperativity. Both inactivation gates appear additively. This may represent a simpler or more parallel style of inactivation mechanism. Because both activation and inactivation are written as sums, $I_{hIR_3}$ can be viewed as the most straightforward extension of the classic HH form, while still allowing multiple gating variables.

The fourth form of eqn (5) is given in eqn (9). Dual gating is introduced by combining two parallel gating pathways – $(m_1, h_1)$ and $(m_2, h_2)$ – *additively* yet retaining a *multiplicative* relationship within each pathway.

$$I_{hIR_4} = g_{hIR} \times \left(m_1{}^{n_4} \times h_1{}^{n_5} + m_2{}^{n_6} \times h_2{}^{n_7}\right)$$
$$\times (V_m - V_r) \tag{9}$$

where the exponents $n_4$ to $n_7$ regulate cooperativity in each distinct gating branch. This structure allows (1) either branch to contribute to the total current independently for reflecting a receptor or channel that owns two separate subunit sets or (2) dual-mode activation/inactivation.

Finally to accommodate the unique gating kinetics of specific purinergic receptors, we introduce a fifth current formulation of eqn (5) given in eqn (10):

$$I_{hIR_5} = g_{hIR} \times m_1{}^{n_4} \times m_2{}^{n_5} \times h_1{}^{n_6} \times h_2{}^{n_7}$$
$$\times (V_m - V_r) \tag{10}$$

This fully multiplicative formulation independently scales contributions from activation $(m_1, m_2)$ and inactivation $(h_1, h_2)$ gates via receptor-specific exponents $(n_4 - n_7)$. Unlike prior additive or hybrid forms, $I_{hIR_5}$ is tailored for receptors where both activation and inactivation pathways exhibit strong cooperativity and non-linear interdependence. The exponents represent subunit stoichiometry or cooperative binding which is required for gating transitions. They enable precise replication of receptor dynamics where complex, interdependent activation and inactivation occur. This arrangement makes the model suitable for capturing receptor dynamics in which separation of gates is required in a completely multiplicative manner.

These five current forms provide a flexible toolkit for modelling diverse ionotropic receptor types. Empirically the best choice (or exponents $n_4$, $n_5$, $n_6$, $n_7$) can be determined by fitting electrophysiological data (e.g. whole-cell or patch-clamp recordings). In some cases it may be necessary to compare the goodness of fit under each formulation to decide which gating structure – additive *vs.* multiplicative activation and inactivation – best reproduces experimental current dynamics. The selection of these forms is discussed in the Results section for different receptors. Ultimately all forms share the same overarching principle: the net current is proportional to a maximal conductance $g_{hIR}$ multiplied by a *gating expression* involving $\{m_1, m_2, h_1, h_2\}$ and the *driving force* $(V_m - V_r)$. By allowing sums or products of the gating variables and by applying integer exponents, each equation can capture the cooperative, parallel or sequential transitions that are characteristic of many human purinergic and glutamatergic receptors.

## Model fitting

As an accurate parameter estimation of the gHH model is a complex task, a non-linear optimisation technique which works in two stages was implemented. Multiple initial guesses are generated randomly for parameters in advance of the first phase; note that this is repeated for each agonist concertation separately. In the first stage, the Levenberg–Marquardt algorithm, which effectively converges to near-optimal solutions, optimises these initial guesses (Moré, 2006). If the outcome of the first stage exceeds the numerical ranges within which parameters are defined, a secondary phase that implements the trust-region-reflective algorithm is initiated (Ashyraliyev et al., 2009; Conn et al., 2000). This will ensure that the resultant optimised parameters are within biological ranges that are defined at the beginning of optimisation. Once these phases get completed, the parameter set that best minimises the error – defined as the goodness-of-fit (sum of squared errors, SSE) between model simulations and experimental data – is selected. This combined approach can help navigate the multidimensional parameter space, such that the final parameters can accurately capture the kinetics observed in the experimental data and reliably predict unseen data. As the exponents $n_1$, $n_2$, …, $n_7$ create non-linear dependencies in the model, it is key how they are determined. In the current implementation integer exponents are chosen from the range [0, 5] such that the best fit of the model to experimental data is obtained. Therefore during optimisation each exponent $n_i$ was constrained to the integer range 0–5. This bracket is wide enough to span plausible stoichiometries (0 = no coupling; 1 = independent; 2–3 ≈ dimer/trimer;

**Table 1. Parameter values for modelling human P2X$_{1-7}$ and hGluA1 receptors: the exponent values were used in eqns (1) through (10), alongside the recovery rate modifier function $\varphi(A)$, and the maximal conductance ($g_{hIR}$) value.**

| Receptor | $n_1$ | $n_2$ | $n_3$ | $n_4$ | $n_5$ | $n_6$ | $n_7$ | $\varphi(A)$ | Current form | $g_{hIR}$ |
|---|---|---|---|---|---|---|---|---|---|---|
| hP2X$_1$ | 2 | 1 | 1 | 1 | 1 | 2 | n/a | 0 | $I_{hIR_2}$ | 136.57 [µS] |
| hP2X$_2$ | 1 | 3 | 1 | 1 | 3 | 1 | 3 | 0 | $I_{hIR_4}$ | 88.92 [µS] |
| hP2X$_3$ | 3 | 1 | 1 | 1 | 1 | 4 | n/a | 0 | $I_{hIR_2}$ | 102.63 [nS] |
| hP2X$_4$ | 3 | 1 | 1 | 5 | 4 | n/a | n/a | 0 | $I_{hIR_3}$ | 7.064 [µS] |
| hP2X$_5$ | 2 | 2 | 2 | 3 | 3 | 3 | n/a | 0 | $I_{hIR_1}$ | 33.87 [nS] |
| hP2X$_6$ | 1 | 1 | 1 | 5 | 2 | 2 | n/a | 0 | $I_{hIR_1}$ | 8.47 [nS] |
| hP2X$_7$ | 2 | 2 | 2 | 3 | 3 | 3 | n/a | 0 | $I_{hIR_1}$ | 33.87 [nS] |
| hGluA1 | 2 | 1 | 1 | 3 | 4 | 3 | 4 | $e^{-A}$ | $I_{hIR_5}$ | 2691.6 [pS] |

*Note*: n/a means that the value in the corresponding cell does not apply to the receptor, which of course depends on the current form used.

4–5 = highly cooperative, akin to the classical 4th-power in the HH K$^+$ model (Hodgkin & Huxley, 1952)), yet narrow enough to prevent over-fitting; thus the $n_4 = 5$ found for hP2X$_4$ signals an especially steep effective cooperativity, not five physical subunits (see Table 1). Note that $g_{hIR}$ is obtained from the maximal current at the highest ligand (ATP or Glu) pulse. Because this conductance is re-estimated for every individual cell (from its own peak current) and then held fixed for that cell's remaining ligand levels, the procedure implicitly captures cell-to-cell variability in receptor expression while still allowing the kinetic rate constants to be fitted globally.

## Results

In this section we show how the gHH model introduced above can accurately capture ion currents of the entire human P2X receptor family as well as the GluA1 receptor. We first fit each receptor's model to its relevant experimental data and show that only one current form out of five equations given in eqns (6)–(10) is appropriate for each receptor within a unified mathematical framework. For each receptor we fitted all five candidate current forms (eqns 6–10) to the full multi-trace data set and retained the form with the lowest SSE; only one form provided a satisfactory fit across all agonist concentrations for that receptor. It is naturally followed that each rate constant is a non-linear function of the agonist concentration. The form of this function can have biological implications in how these rates regulate different dynamics such as channel activation, inactivation and recovery. Each model is then integrated numerically for two goals: (1) making predictions about the time course of whole-cell currents and (2) gaining insight into each receptor's gating properties. We also confirmed that removing activation gate ($m_2$) and inactivation gate ($h_2$) degraded the fit markedly (see Supporting Information S1.3, Fig.

S1.11), indicating that the four-gate architecture is the minimal structure that reproduces all 38 human P2X$_{1-7}$ and GluA1 recordings. The models were implemented in MATLAB Release 2024b using three major MATLAB routines (the model code is publicly accessible as given in Supporting Information S3). Although the model differential equations were non-linear and stiff, the *ode15s* solver successfully provided the best stability in numerical integration. The curve fitting component, as discussed earlier, leveraged two non-linear least-squares (NLS) routines from the MATLAB optimisation toolbox, namely, trust-region-reflective and Levenberg–Marquardt algorithms. The results demonstrated that the integrated use of these two powerful algorithms can efficiently explore the high-dimensional parameter space of the gHH model. Additionally the experimental current traces were extracted by using WebPlotDigitizer from published figures. To access original data files, see Supporting Information S2, which includes all digested datasets with units and experimental conditions. What is followed below are the fittings, simulations and sensitivity analysis for the human P2X$_{1-7}$ and GluA1 receptors.

### Human P2X$_1$ model

The human P2X$_1$ contributes to various physiological processes including smooth muscle contraction in arteries, bladder and vas deferens (Lee et al., 2000; Vulchanova et al., 1996), enteric nervous system activity (Ji et al., 2018), endocrine signalling in pancreatic islets (Ji et al., 2018) and platelet function (Bennetts et al., 2022). We took a robust dataset that supports a diverse range of ATP levels to construct the hP2X$_1$ model (Roberts & Evans, 2004). To model the hP2X$_1$R we have identified that the second variant of the current form (eqn (7)) can significantly capture the receptor dynamics best in comparison to the other forms whose exponents are listed in Table 1.

Responses of the fitted model appear in Fig. 2 across five ATP concentrations from 0.1 to 100 µM. The selection of the specific exponents and the current form validated that the model can precisely capture the activation and inactivation processes which are essential for the function of the hP2X$_1$ receptor. Numerical values for the fitted parameters come in Table S1. 1.

Figure 3 quantifies how the eight biochemical rate constants ($\alpha_{m_1}$, $\beta_{m_1}$, $\alpha_{m_2}$, $\beta_{m_2}$, $\alpha_{h_1}$, $\beta_{h_1}$, $\alpha_{h_2}$, $\beta_{h_2}$) of the hP2X$_1$ model depend on varying levels of ATP concentration. It is necessary to note that such analyses are important to understand the receptor's pharmacodynamics and its sensitivity to physiological and pharmacological variations in ATP. Each subplot within Fig. 3 represents a different kinetic parameter plotted against ATP concentrations ranging from 0.1 to 100 µM, which are normalised to the maximum concentration of 100 µM tested. The rate constants are also normalised to their maximum values derived from the fitting process to ensure a uniform scale for comparison. A piecewise cubic interpolation was applied to the rate constant data, allowing for a smooth visual representation of how non-linearly each parameter varies with ATP concentration.

As ATP increases, the rate constants for activation gates ($\alpha_{m_1}$, $\beta_{m_1}$, $\alpha_{m_2}$, $\beta_{m_2}$, $\alpha_{m_1}$) exhibit two contrasting pathways. $\alpha_{m_1}$ starts from a high value at low ATP and progressively becomes smaller while ATP increases. This suggests that this gate saturates at stronger stimulation by ATP while $m_1$ is quickly activated at low ATP. On the contrary $\alpha_{m_2}$ takes a reverse trend where it remains low at low ATP and goes up as ATP increases. So $m_2$ gate requires more ATP to effectively open. The deactivation rates experience a complex change where $\beta_{m_1}$ starts high and then decreases but again it increases with ATP. However $\beta_{m_2}$ starts low and peaks at $ATP = 1\ \mu M$ and then falls off with the increase in ATP. This hints at a range of ATP during which the $m_2$ gate closes more quickly at mid-levels of ATP. The peak in $\beta_{m_2}$ at 1 µM ATP likely marks a shift between high- and low-affinity de-activation: partial occupancy favours closure, while both low and high ATP stabilise alternative conformations, lowering $\beta_{m_2}$ (Mansoor et al., 2016; Wang et al., 2018). The inactivation gate $h_1$ shows a comparatively simple trend: both $\alpha_{h_1}$ and $\beta_{h_1}$ increase while ATP progresses to higher values. Note that at high levels of ATP, these rates take a very high value. This indicates that after a certain threshold $\beta_{h_1}$ experiences a strong desensitising influence from ATP. $h_2$ inactivation ($\beta_{h_2}$) follows a gradual decrease and finally it stabilises, which hints at $h_2$ gate to become less active as ATP increases. However $\alpha_{h_2}$ shows a distinct sensitivity pattern in which it starts low and peaks at $ATP = 0.3\ \mu M$ then stabilises at a low value. The de-inactivation rates ($\alpha_{h_1}$, $\alpha_{h_2}$) are indicative of how each gate has its own ATP-dependent 'sweet spot' for either promoting or reversing inactivation.

Such detailed mapping for rate constants across a wide range of ATP concentrations lays a foundation for understanding how each receptor plays its role in physiological conditions and what happens when the receptor is dysregulated in pathological states.

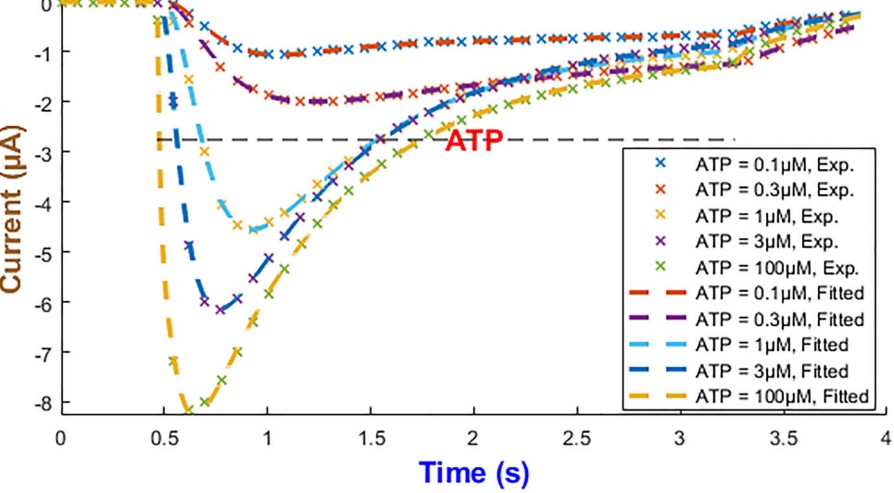

**Figure 2. Fitted current transients for the hP2X$_1$ model across various ATP concentrations**
Experimental (Exp.) and fitted (Fitted) current traces for the human P2X$_1$ receptor are shown over a 4-s time span upon application of ATP at concentrations ranging from 0.1 to 100 µM. Each line represents the average current response measured experimentally (solid lines) (Roberts & Evans, 2004) and the currents predicted by our generalised Hodgkin–Huxley model (dotted lines). In close agreement between the experimental data and the fitted model, both activation and subsequent inactivation phases are activated. The horizontal bar shows the duration of agonist exposures.

Figure 4 quantifies the detailed kinetics through numerical simulation for the hP2X$_1$ receptor by ATP concentration which changes from 0.1 to 120 μM. Two applications of ATP are examined: long term within 26 s in Panel (a), and short term within 3 s in Panel b. The receptor exhibits a gradually increasing current peaking at lower ATP levels (0.1–1 μM) which is followed by a slower decline towards a persistent current. A quicker activation

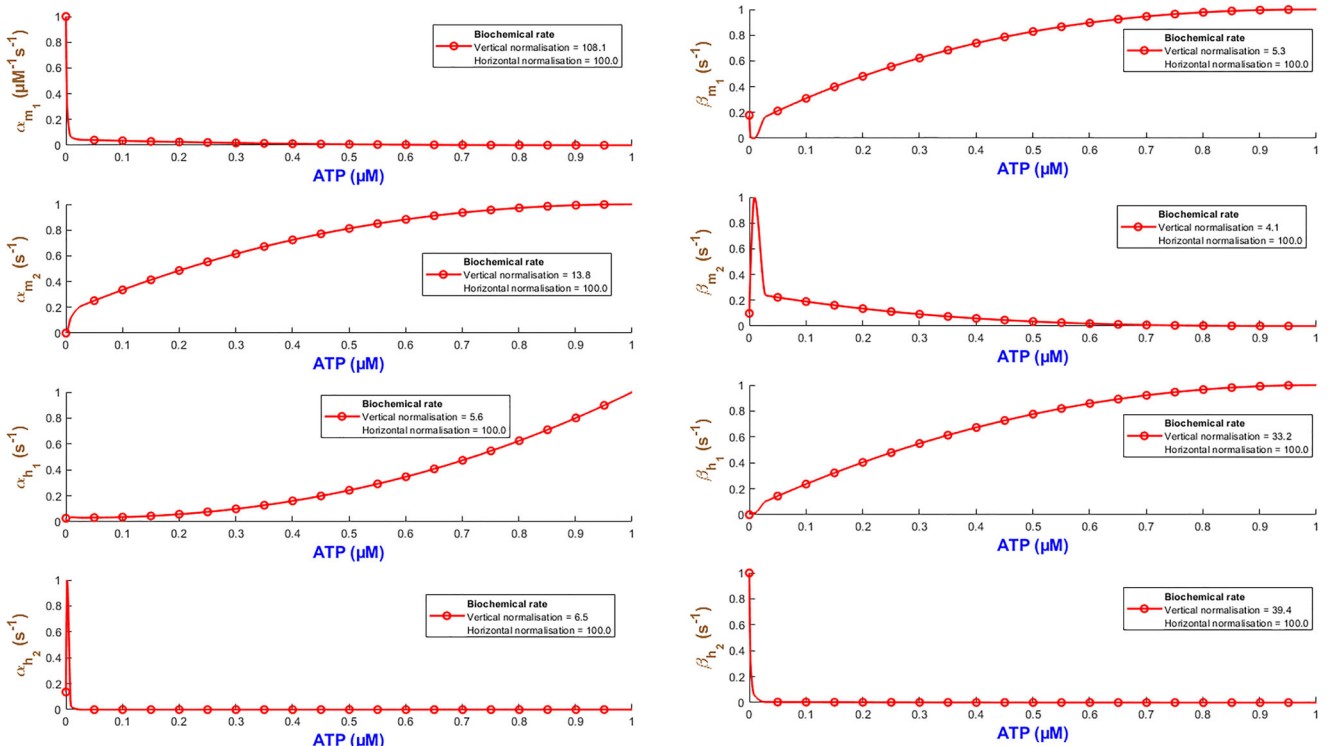

**Figure 3. Normalised biochemical rate constants as functions of ATP concentration**
This figure illustrates the fitted biochemical rate constants ($\alpha_{m_1}$, $\beta_{m_1}$, $\alpha_{m_2}$, $\beta_{m_2}$, $\alpha_{h_1}$, $\beta_{h_1}$, $\alpha_{h_2}$, $\beta_{h_2}$) for the human P2X$_1$ receptor model, each plotted against ATP concentrations normalised to the maximum tested value of 100 μM. Each plot shows how specific rate constants, normalised to their respective maximum values obtained from the fitting process, vary with increasing ATP concentration. A piecewise cubic interpolation has been fitted to these parameters, extracted from fitting eqns (1)–(4) and (6) to experimental data, enhancing the visualisation and analysis of parameter trends across varying agonist levels. The normalisation factors for both ATP concentration (horizontal axis) and rate constants (vertical axis) are specified in each subplot, providing a clear comparative analysis across different kinetic parameters.

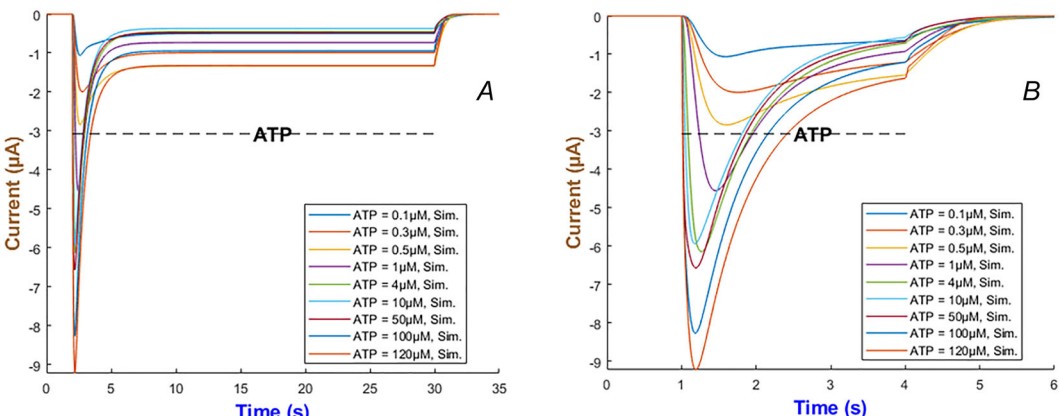

**Figure 4. Simulation of hP2X$_1$R currents**
The left (*A* and *B*) depicts the long-term (28 s) and short-term (3 s) exposure to ATP ranging from 0.1 to 120 μM. The dashed line indicates ATP duration.

is observable when ATP increases (i.e. between 4 and 120 μM), peaking higher and then stabilising at a low baseline despite the prolonged exposure to agonist. A saturating response or an activation balancing with desensitisation can possibly lead to this behaviour. Currents become more sensitive to ATP over the short-term ATP stimulation as the higher ATP concentrations can cause significantly faster responses. In some cells quick responses are necessary (e.g. in synaptic transmission), and these predictions provide insights into how $hP2X_1$ can contribute to such quick kinetics.

We now turn our attention to analysing the dynamics of four mode state variables $m_1$, $m_2$, $h_1$ and $h_2$ in Fig. 5 under a sustained stimulus of 10 μM ATP over 28 s studied in Fig. 4. Within a few seconds of ATP exposure, both $m_1$ and $m_2$ gates rapidly reach their near maximum probabilities and then stabilise to efficiently initiate ion flow across the plasma membrane. The $h_2$ gate displays a relatively slow decline which can prevent the $hP2X_1R$ activity resulting in overexcitation. The $h_1$ gate undergoes a sharp decrease after which both $h_1/h_2$ gates plateau while ATP is present. The time courses of these gates which contribute to modulating the receptor sensitivity function hint at the fine-tuned receptor capacity under prolonged ATP stimulation. These simulations illustrate how seamlessly the $hP2X_1$ receptor handles both rapid and sustained signal transductions in a complex, ATP-dependent manner. Such

kinetics of activation/inactivation channel gating can be applied to guide therapeutic interventions that target the $hP2X_1$ receptor in response to varying levels of ATP levels.

### Human P2X₂ model

The human $P2X_2$ receptor ($hP2X_2$) plays key roles in mediating fast synaptic transmission among both central and peripheral neurons (Galligan & North, 2004; Ren et al., 2003), driving reactive enteric gliosis (Schneider et al., 2021), and contributing to diverse physiological processes, including pain perception and neuromodulation (Burnstock, 2014; Khakh, 2006). To examine the ATP-dependent gating characteristics of $hP2X_2$ we used published current recordings from the wild-type receptor over a wide range of ATP concentrations (Roberts et al., 2008). This dataset is an ideal benchmark to illustrate our gHH precision to capture the rapid activation and partial desensitisation observed in its current traces.

In modelling the $hP2X_2R$ we specifically employed the *fourth current form* in eqn (9), because fittings showed that other forms (eqns 6–8) could not reproduce the pronounced dip and subsequent recovery at varying ATP levels. Eqn (9) incorporates dual gating in additive form with integer exponents $\{n_4, n_5, n_6, n_7\}$ chosen from the discrete set $\{0,...,5\}$ to best fit the data. As summarised in Table 1, the final exponent values

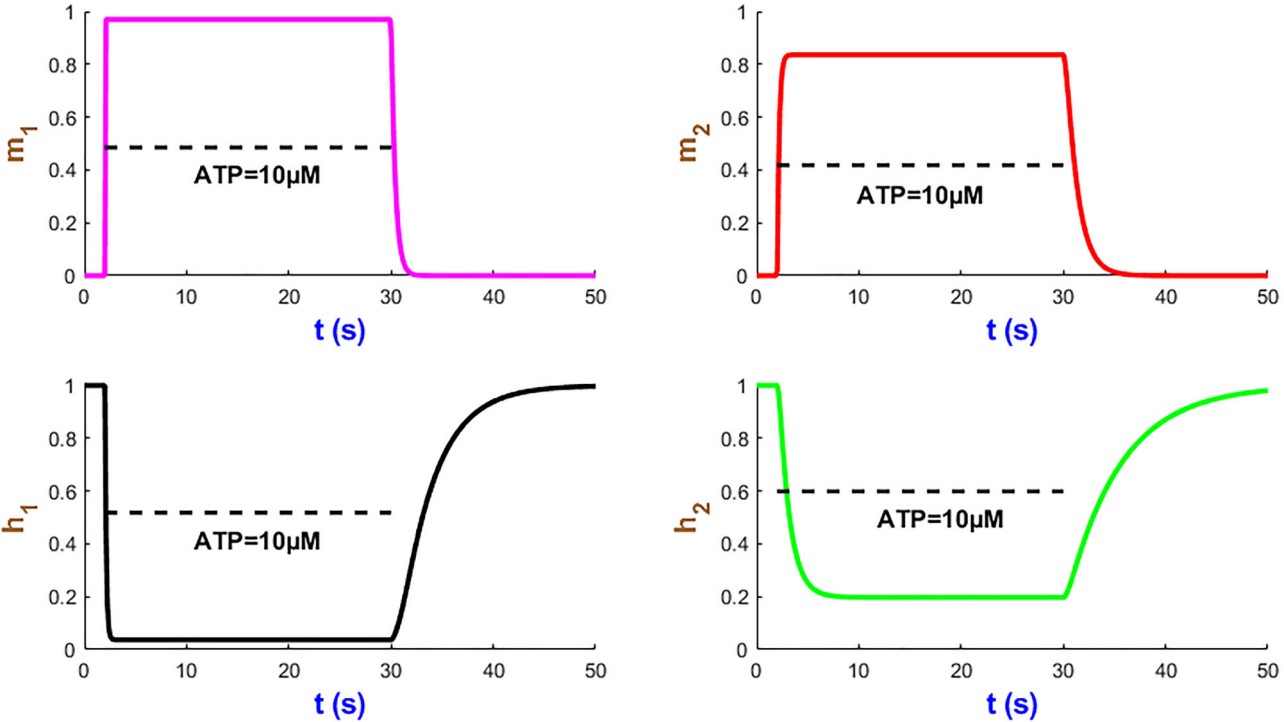

**Figure 5. Simulation of hP2X₁ state variables at AT*P* = 10 μM**
This figure showcases the time course of the state variables $m_1$, $m_2$, $h_1$ and $h_2$.

and maximal conductance effectively capture the dual activation–inactivation dynamics across all tested ATP concentrations. As portrayed in Fig. 6, the model predictions agree closely with the experimentally recorded currents after curve fitting the model to find the best rate constants and using $I_{hIR_4}$ across five levels of ATP. All optimised parameters are reported in Table S1. 3 and visualised in Fig. 7, which are functions of ATP concentration.

Most of the rate constants for hP2X$_2$R do not behave monotonically with an increase in ATP in Fig. 7. The rate constant $\alpha_{m_1}$ is small at low levels of ATP concentrations. It increases at intermediate levels and then decreases again as ATP becomes large. Moderate ATP concentrations strongly promote the forward (activation) transition of the m$_1$ gate, but at very high agonist levels, other gating processes (e.g. partial inactivation or alternative subunit transitions) can counteract further increases in $\alpha_{m_1}$. Some rates decrease steadily with increasing ATP (e.g. $\beta_{m_1}$). However plateauing behaviour or inflection points are observed at some other rates. The channel most efficiently transitions into open states at certain levels of ATP, whereas inactivation or subunit heterogeneity can dominate at higher or lower levels. These biphasic changes which are highly ATP-dependent suggest multiple energetic barriers (such as multiple conformational or binding steps), affinity states or competing pathways that finely tune the hP2X$_2$R gating properties, which our gHH model captures simply and universally. It is worth noting that the maximum conductance g$_{hIR}$ is kept fixed during the fittings. Instead it

is the rate constants that absorb concentration-dependent behaviour.

Figure 8 demonstrates the simulation of the hP2X$_2$ model under the same duration in ATP application but with concentrations ranging from 1 to 300 μM. Panel (A) – long ATP exposure – shows the current quickly rises to a near-peak level and then partially inactivates consistent with experimental evidence (North, 2002). Sustained application of ATP prolongs the open state so that the current remains significantly above baseline for the entire stimulation window. In short ATP application – Panel (B) – the hP2X$_2$R quickly recover once ATP is removed. Higher ATP concentrations induce a larger peak current and more inactivation. The overall transient after agonist removal is still dictated by the rate constants that govern each gating transition.

The dual branches associated with the receptor current in eqn (9) along with the rate constants which are concentration dependent allow for rapid activation at both low and high ATP levels to capture the subtle interplay of partial inactivation and deactivation. Despite spanning three orders of magnitude in agonist concentration, the multifaceted gating kinetics can be captured by the gHH formalism accurately as confirmed by the predictions.

The temporal evolution of the gating variables of the hP2X$_2$ receptor is illustrated in Fig. 9. Activation gates $m_1$ and $m_2$ exhibit rapid and slow initial activation ($t < 5s$), respectively, as ATP binds $m_1$ and promotes channel opening. However inactivation gates $h_1$ and $h_2$ decline gradually to allow the receptor to partially desensitise during sustained agonist stimulation. The

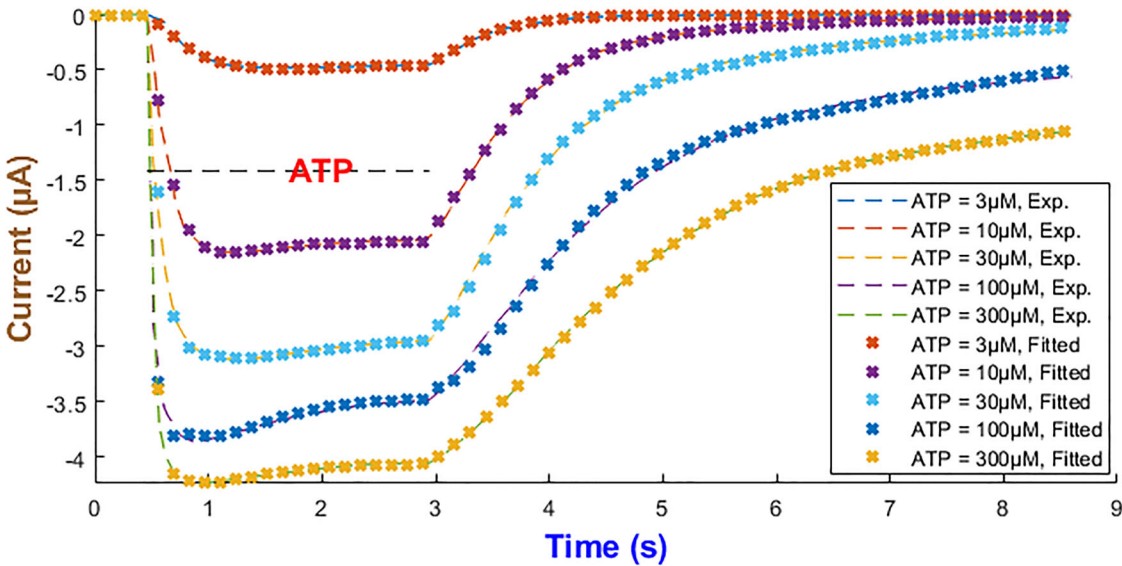

**Figure 6. Model fidelity across ATP concentrations for the hP2X$_2$ Receptor**
This figure illustrates the precision of the fourth form of our generalised Hodgkin–Huxley (HH) model (eqn (9)) in replicating the experimental current dynamics of the hP2X$_2$ receptor across ATP concentrations ranging from 3 to 300 μM (Roberts et al., 2008).

biphasic interplay of activation/inactivation gates shapes the hP2X$_2$R's ability to balance rapid signal transduction with self-regulation. The cooperative gating and agonist-dependent modulation intrinsic to P2X$_2$ receptors is also affected by the dual pathway of $I_{hIR_4}$ current.

More importantly the value of integer exponents ($n_1 = 1$, $n_4 = 1$) and ($n_2 = 3$, $n_5 = 3$) that modulate the $m_1$ gate further aligns with the trimeric stoichiometry known to hP2XRs. Non-linear inactivation is dictated by sub-

unit interactions. These results show that the model can describe macroscopic currents whose behaviour is regulated by the underlying molecular-scale regulatory mechanisms.

## Human P2X$_3$ model

The hP2X$_3$ receptor is highly expressed in sensory and enteric ganglia and contributes to pain perception,

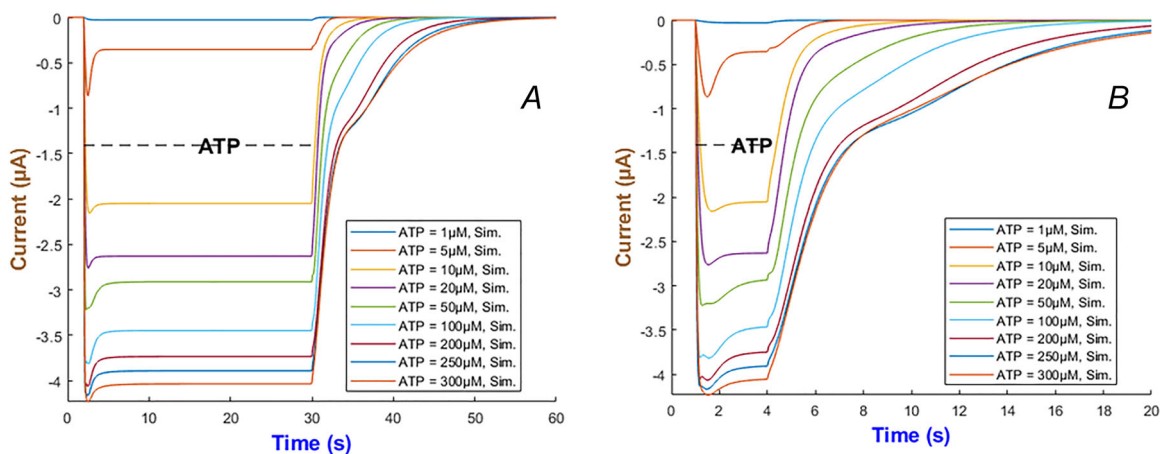

**Figure 7. Normalised rate constants of the hP2X$_2$R with respect to ATP**
All values are normalised according to the maximum rates specified in Fig. 6.

**Figure 8. Simulation of hP2X$_2$ currents to varied ATP durations**
*A* and *B*, illustrate the simulated responses to long and short applications of ATP across concentrations ranging from 1 to 300 μM.

mechanosensation and gut motor and secretory control (Burnstock, 2000; Galligan, 2004). Apart from other P2X receptors, $hP2X_3R$ exhibits a long recovery phase and plays a major role in chronic pain disorders. Therefore the development of potential therapeutic targets requires a complete understanding of its gating properties (Giniatullin & Nistri, 2013). To effectively capture three major $hP2X_3$ phases of the activation, desensitisation and recovery it was found that $I_{hIR_2}$ is the best choice out of the five current forms as depicted in Fig. 10. The experimental datasets for model fitting come from Pratt et al. (2005) and Riedel et al. (2012). The best model exponents along

with the maximal conductance of the $hP2X_3$ model are summarised in Table 1.

The kinetic rate constants with respect to ATP for the $hP2X_3R$ model are given in Fig. 11. The fitted numerical values of these rates are also detailed in Table S1. 5. The activation/deactivation rates disclose complex gating interactions in $hP2X_3$ receptors.

$\alpha_{m_1}$ decreases monotonically. The rate $\alpha_{m_2}$ starts by falling into a valley at ATP = 0.3 μM before rising sharply at high ATP. It finally increases and peaks at ATP = 100 μM. This means that $m_2$ activation is a secondary gating process that becomes more pronounced

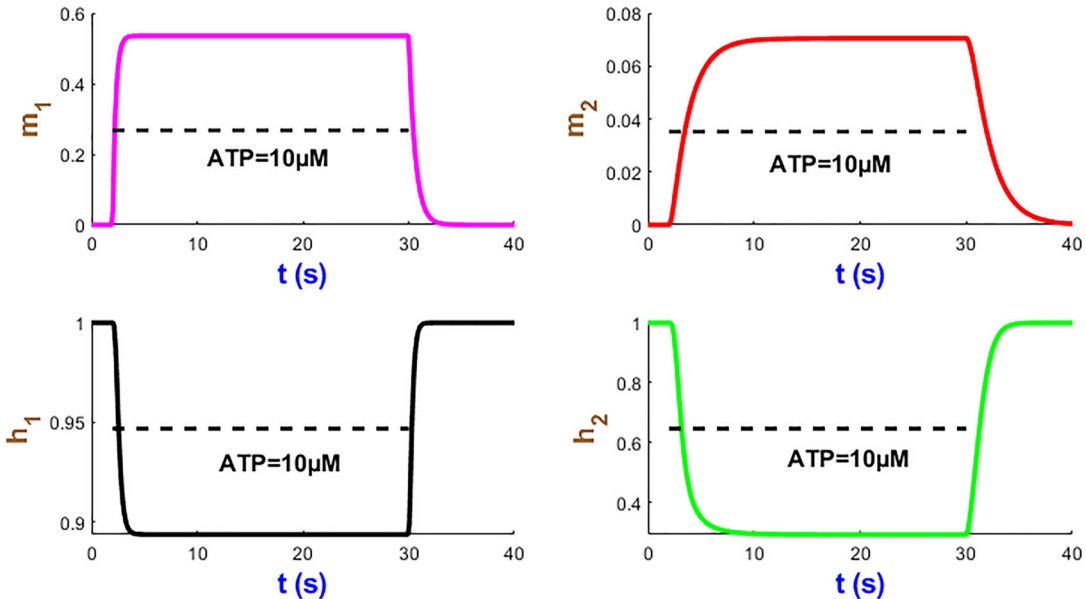

**Figure 9. Simulated dynamics of the $hP2X_2$ gating variables at 10 μM ATP**
This figure displays the temporal profiles of the gating variables $m_1$, $m_2$, $h_1$ and $h_2$ of the $hP2X_2$ receptor during a 30-s stimulation.

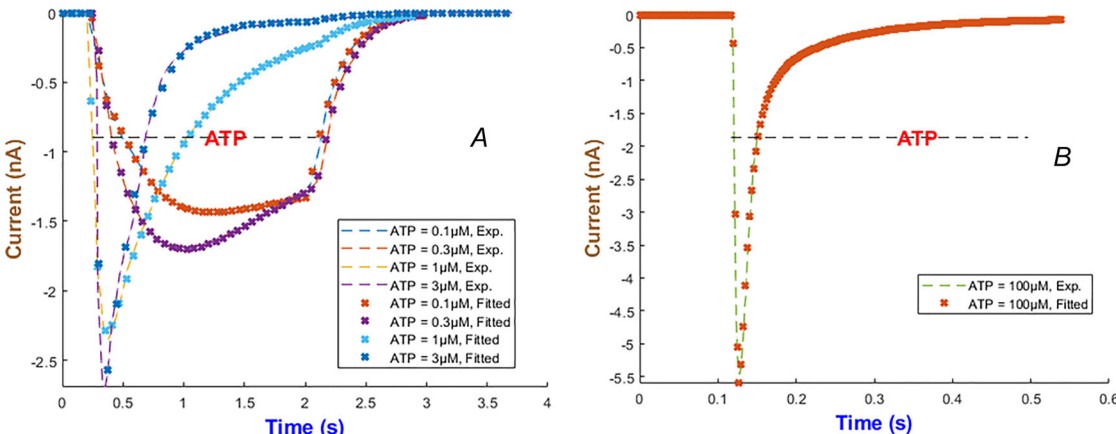

**Figure 10. Model fidelity across ATP concentrations for the $hP2X_3$ receptor**
This figure illustrates the precision of the second form of our generalised Hodgkin–Huxley (gHH) model (eqn (7)) in replicating the current dynamics of the $hP2X_3$ receptor across low and high levels of ATP concentrations ranging from 0.1 to 3 μM (Panel A) and 100 μM (Panel B) (Pratt et al., 2005; Riedel et al., 2012).

at high ATP levels potentially reflecting cooperative binding controlled by the $m_1$ gate. At low levels of ATP, the $\beta_{m_1}$ rate remains relatively stable but goes to a slight minimum at ATP = 3 μM followed by a gradual increase. A strongly increasing trend is observable for $\beta_{m_2}$. What this means is that the $hPX_3R$ is optimised for rapid activation at low ATP and de-activates more rapidly at higher values of ATP to limit prolonged channel opening and is likely to prevent excessive ion flux.

The desensitisation/recovery in $hP2X_3$ receptors is regulated by inactivation/de-inactivation rates, where $\beta_{h_1}$ and $\alpha_{h_1}$ behave intricately. $\beta_{h_1}$ first peak at ATP = 0.1 μM. It then falls into a valley at ATP = 3 μM. From there it rises sharply again at ATP = 100 μM. So inactivation is strong at low and high levels of ATP but acts weakly at other (intermediate) levels. This can potentially happen when the receptor creates an ATP-dependent shift in its gating conformation.

$\beta_{h_2}$ after a few oscillations increases supporting the idea that higher ATP levels promote deeper desensitisation. $\alpha_{h_1}$ exhibits a local peak at 0.3 μM ATP but declines and reaches a low baseline which continues to ATP = 100 μM. These together manifest the $hP2X_3R$ ability, respectively, in efficient recovery from inactivation at low ATP and slow recovery at higher ATP levels leading to prolonged desensitisation. $\alpha_{h_2}$ tends to follow a generally increasing

trend from ATP = 3 μM to ATP = 100 μM with no oscillatory behaviour.

The multi-scale temporal dynamics of the $hP2X_3$ receptor in Fig. 12 reveals distinct timescales of activation, desensitisation and recovery with respect to ATP concentration. At nanomolar ATP levels (i.e. <1 μM), as seen in Fig. 12*A* and *B*, receptor desensitisation occurs on the order of seconds, allowing currents to persist throughout the short (2-s) and long (28-s) ATP exposure. On the opposite side it desensitises in milli-seconds at higher levels of agonist. The inactivation gate $h_2$ exhibits a monophasic recovery at low ATP levels as shown in Fig. 12*C* and *D*. But interestingly it transitions to a biphasic mode at ATP = 100 μM. This can be caused potentially by a possible shift in receptor conformational dynamics which is in turn influenced by ATP binding. This transition happens over an intermediate timescale of tens of seconds due to ligand-induced stabilisation of desensitised states. At the longest timescale, receptor recovery spans from seconds to minutes depending on ATP concentration which prolongs receptor unavailability.

Figure 13 presents the predicted full recovery time of $hP2X_3$ with respect to ATP concentration. The recovery times plateaus at higher levels of ATP in a monotonically increasing form from low levels. The receptor takes

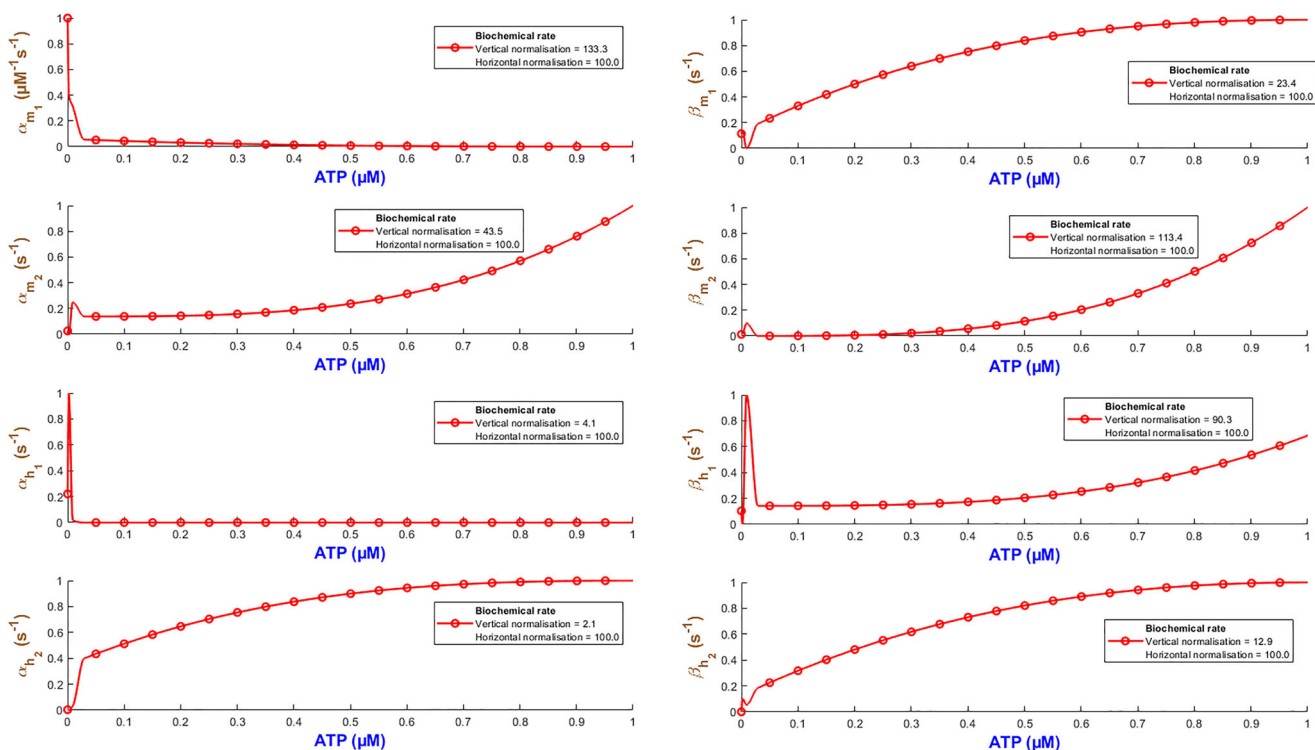

**Figure 11. Fitted and normalised rate constants for the $hP2X_3$ receptor with respect to ATP**
All values are normalised according to the maximum rates specified in Fig. 10.

about 8 min to fully recover from its desensitisation state at ATP = 30 μM. This is confirmed by the work in Pratt et al. (2005), who recorded hP2X$_3$ receptors expressed in HEK293 cells and found that a saturating 30 μM ATP pulse required approximately 8 min for full recovery from desensitisation. Stronger ligand-receptor interactions, slower unbinding and conformational transitions can stabilise the receptor in a desensitised state and limit further recovery delays. Additionally this observed saturation may be affected by (unknown) intracellular regulatory mechanisms that finely tune the hP2X$_3$ receptor availability when ATP concentration changes.

The gHH model integrates sub-second activation, intermediate-scale gating transitions and long-term recovery within a single mathematical structure. This capability makes the model a powerful tool for understanding how ligand-gated receptors such as P2XRs work across multiple temporal regimes by offering key mechanistic insights into ATP-mediated regulation.

## Human P2X$_4$ model

The hP2X$_4$ receptor plays a critical role in immune responses, the regulation of neuroinflammation (Sophocleous et al., 2022) and CNS homeostasis and pathology associated with microglia. It has unique sensitivity to modulatory factors which makes its gating properties complex. A quantitative characterisation of this receptor could help improve therapeutics for neurological disorders like Alzheimer's disease (Castillo et al., 2022). To adapt the gHH framework to the human P2X$_4$ receptor, $I_{hIR_3}$ was chosen thanks to its additive contributions from activation ($m_1 + m_2$) and inactivation ($h_1 + h_2$) gates. As listed in Table 1, the fitted exponents $n_4 = 5$ and $n_5 = 4$ indicate a strong cooperative gating by introducing a strong non-linear dependence on activation and inactivation gates. Human data taken from Ilyaskin et al. (2019) were used to validate the model in Fig. 14.

Figure 15 illustrates the hP2X$_4$ model's rate constants with respect to ATP (also see Table S1. 7). Activation rates behave biphasically. $\alpha_{m_1}$ first rises to a peak at inter-

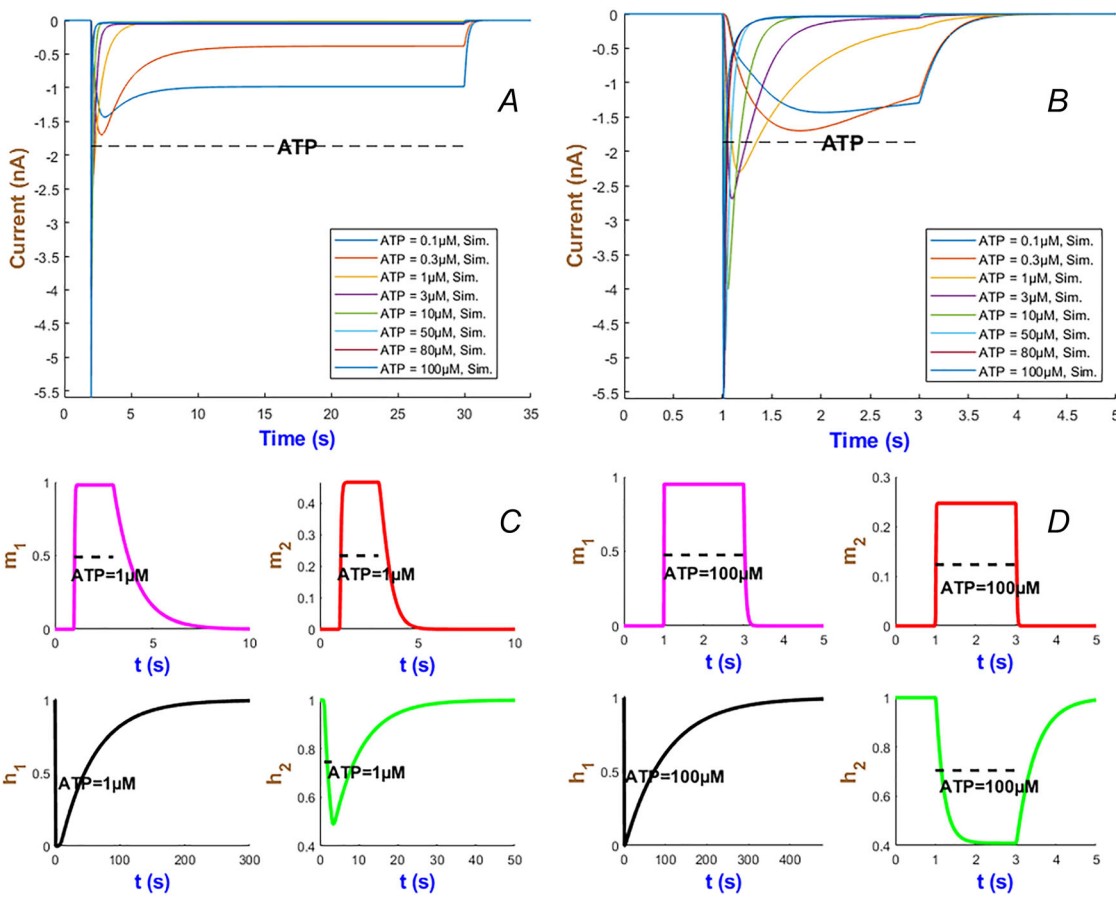

**Figure 12. Simulation of currents and gating variables for the hP2X$_3$R model**
*A* and *B*, illustrate the responses of the hP2X$_3$ receptor to long and short applications of ATP across concentrations ranging from 0.1 to 100 μM. *C* and *D*, display the temporal profiles of the gating variables $m_1$, $m_2$, $h_1$ and $h_2$ of the model during a 2-s stimulation with 1 and 100 μM ATP. Responses fully recover within 5 min at ATP = 1 μM and within 8 min at ATP = 100 μM.

mediate levels of ATP before declining. $\alpha_{m_2}$ first falls into a valley and then rebounds slightly before saturating. Both deactivation rates monotonically decrease when ATP is low. Then they reach stable plateaus at higher concentrations under progressive desensitisation. $\alpha_{h_1}/\alpha_{h_2}$ rates change in three phases. The triphasic behaviour of $\beta_{h_2}$ appears in $\alpha_{h_2}$. But $\beta_{h_1}$ tends to be monotonic and stabilises at a low-level baseline.

Simulations of the hP2X$_4$ model appear in Fig. 16 to ATP applications with long and short durations. Currents in Panel (A) in Fig. 16 reach their steady-state activation within 5–8 s. This is followed by gradual inactivation which mirrors experimental desensitisations in Ilyaskin et al. (2019). Figure 16*B* shows, in brief ATP pulses, currents rapidly are activated ($\tau_{act} < 1\ s$) and then recovered immediately. Both are consistent with

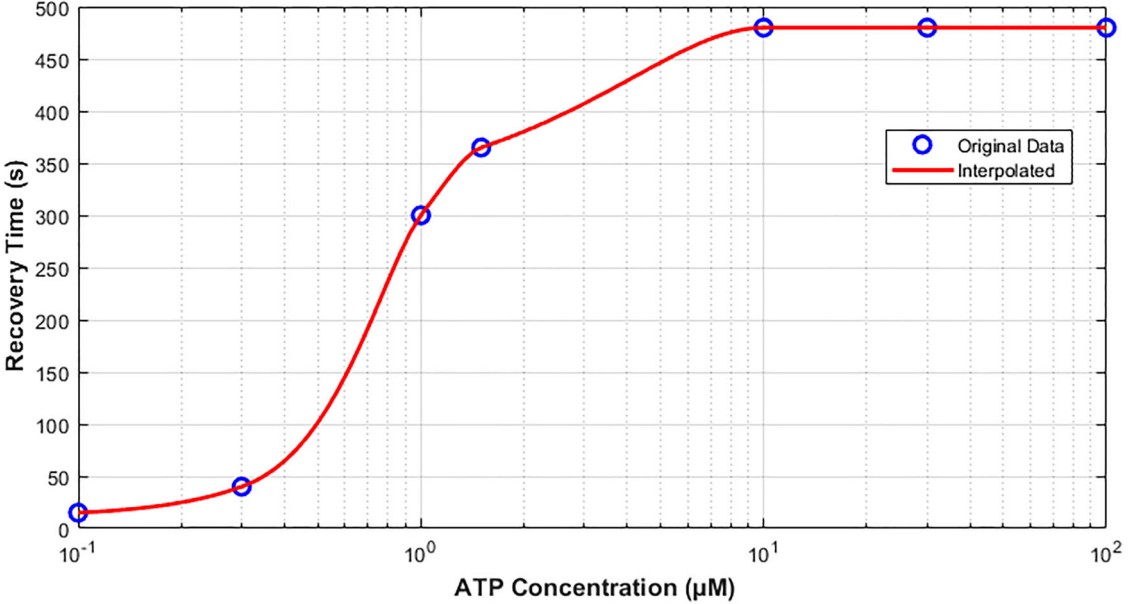

**Figure 13. Full recovery time of hP2X$_3$R as a function of ATP**
This plot illustrates the predicted full recovery time of the human P2X$_3$ receptor as a function of ATP concentration, estimated for a 2-s ATP application. The *x*-axis is based on logarithmic scale. A steep rise is seen in recovery time around ATP = 1 μM and plateaus beyond ATP = 3 μM. For predicted recovery times of hP2X$_1$ and hP2X$_2$ receptors, see Supporting Information S1. 4.

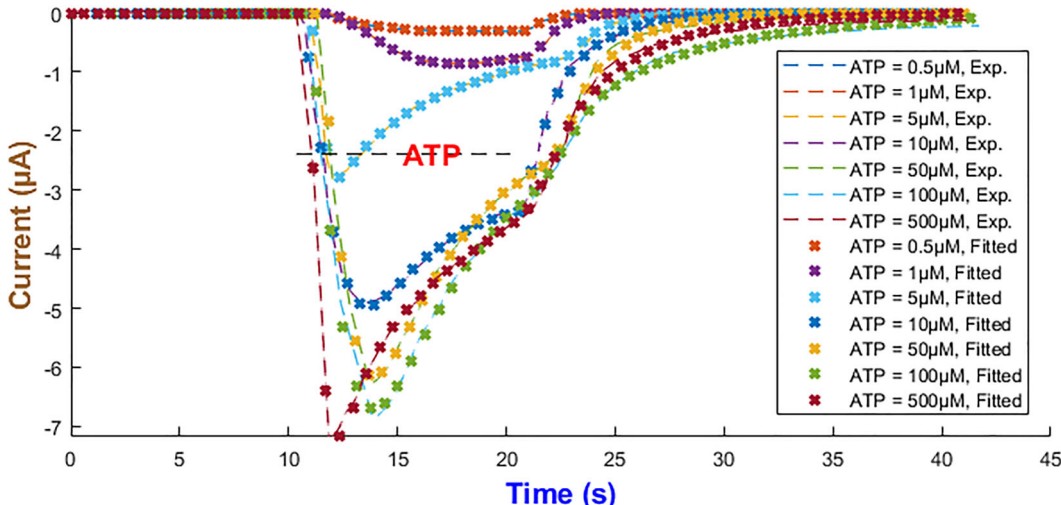

**Figure 14. Fitted current transients for hP2X$_4$ model across a wide range of ATP levels**
$I_{hIR_3}$ closely captures the current dynamics of the hP2X$_4$ receptor experimentally reported in Ilyaskin et al. (2019).

physiological ATP transients in synaptic transmission or paracrine signalling. The accuracy of these predictions makes the model suitable for simulating tonic and phasic signalling, whether in hepatic ATP release or immune cell activation.

Figure 17 shows the simulation of the hP2X$_4$ model's gates. $m_1$ increases much faster than $m_2$ before both statures. $h_2$ declines slower than $m_2$'s rise. The rapid decay of $h_1$ and rapid rise of $m_1$ correspond to each other. The biphasic nature of the hP2X$_4$ responses is emulated using the additive form of activation and inactivation gates in $I_{hIR_3}$ to support cooperative channel gating. This not only simplifies the model development within a single mathematical structure but also preserves the hP2XRs' biophysical fidelity.

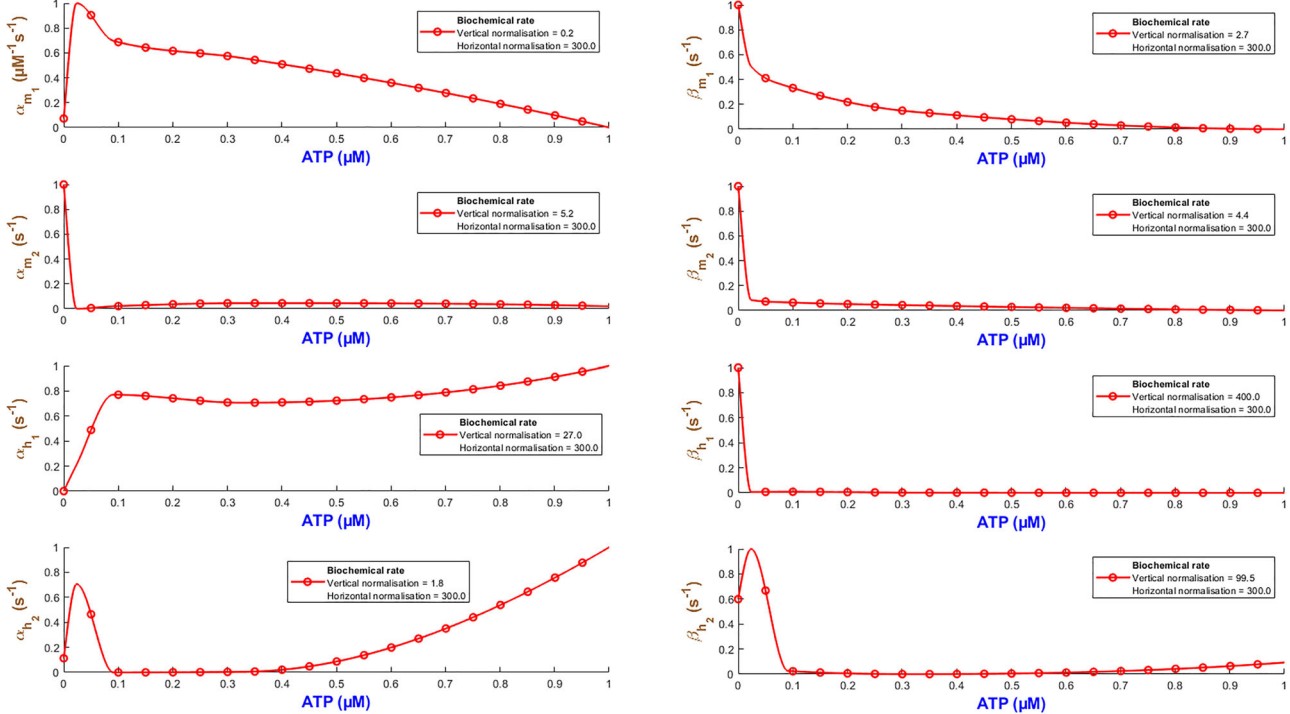

**Figure 15. Fitted and normalised rate constants for the hP2X$_4$ receptor across a wide range of ATP concentrations**
Note that values were normalised to the peak rates introduced in Fig. 14.

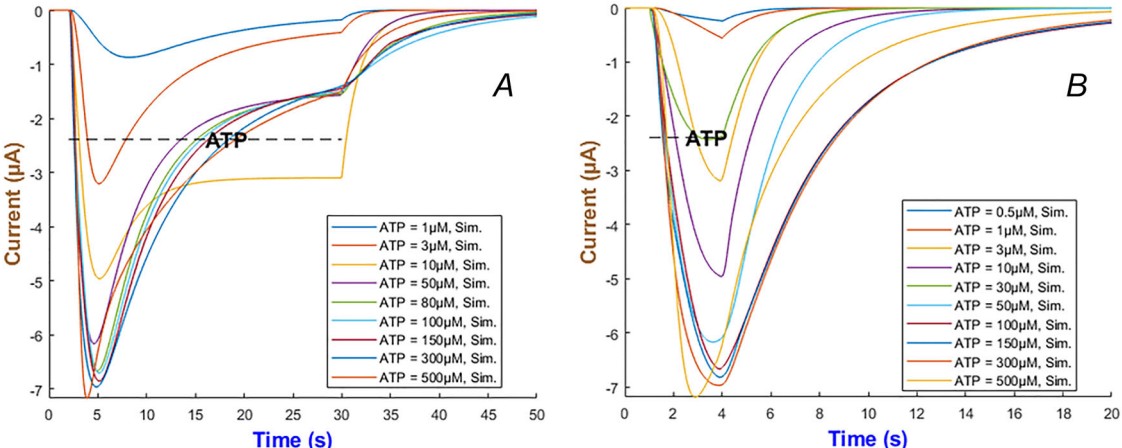

**Figure 16. Simulation of the hP2X$_4$ model across a wide range of ATP levels**
*A* and *B*, display the hP2X$_4$ receptor's responses over extended (28 s) and brief (3 s) ATP exposure periods, respectively.

## Human P2X$_5$ model

The hP2X$_5$ receptor functions as an important ion channel in skeletal (Kim et al., 2017), immune, neural (Dunn et al., 2001; Ruan & Burnstock, 2005) and cardiac cells (King, 2023) and is permeable to large cations (e.g. NMDG) and chloride ions. In order for the gHH model to capture the human P2X$_5$'s dataset reported in Bo et al. (2003), we adopted $I_{hIR_1}$ to integrate additive activation with multiplicative inactivation. Fitted exponents and rate constants are listed in Table 1 and Table S1. 9, respectively. Because the fitted exponents are all in orders of either two or three, cooperative ATP binding and subunit interactions strongly regulate the receptor consistent with its trimeric stoichiometry. The model is capable of closely replicating ATP-gated activation with partial desensitisation for ATP = 1–100 μM (see Fig. 18).

hP2X$_5$'s fitted parameters are remarkably more complex than hP2X$_{1-4}$ receptors (see Fig. 19). $\alpha_{m_1}$ starts declining at low values of ATP forms a valley, peaks at ATP = 10 μM and then decreases exponentially. $\beta_{m_1}$ goes up to a peak at ATP = 3 μM, collapses sharply at ATP = 10 μM and slowly rebounds at higher ATP. Both $\alpha_{m_2}$ and $\beta_{m_2}$ peak at ATP = 30 μM and then decay through a synchronised subunit activation, which is also mirrored by $\alpha_{h_1}$. But $\beta_{h_1}$ decays from a high value and shows a late-phase uptick. $\alpha_{h_2}$ peaks at ATP = 10 μM, plunges to a valley, then rises.

Figure 20 predicts how hP2X$_5$R responds to agonists. The receptor maintains a baseline after peaking and

gradual desensitisation (North, 2002) when ATP is high. In lower levels of agonist, the receptor is rapidly activated ($\tau_{act} < 1\ s$) and fully recovers upon agonist removal. This detailed information is useful to understand scenarios in which tonic (e.g. muscle contraction) and phasic (e.g. immune cell responses) signalling is mediated by P2X$_5$ receptors (Khakh, 2006).

To investigate the hP2X$_5$'s kinetic details, the time course of its four gates is drawn in Fig. 21 through a stimulation of ATP = 10 μM. All the three gates $m_1$, $m_2$ and $h_1$ exhibit similar biphasic responses. $m_2$ plateaus at a much lower level than $m_1$, and their difference directly contributes to the additive term $(m_1 + m_2)^{n_4}$ in $I_{hIR_1}$ which itself describes sequential subunit cooperation. In comparison to all other gates, $h_2$ decays monotonically while ATP is in progress and no plateau appears, which is unique to the hP2X$_5$ model.

## Human P2X$_6$ model

The hP2X$_6$ receptor is widely distributed throughout the central (Collo et al., 1996) and enteric nervous systems (Yu et al., 2010) and regulates ATP-mediated synaptic transmission. The gHH model was applied to the only available human P2X$_6$ dataset (Collo et al., 1996), which contains a single level of ATP application. The quality of the fit is shown in Fig. 22*A*. Numerical values of fitted rate constants and model exponents are given in Tables S1. 11 and 1. The model captures the slow activation and delayed

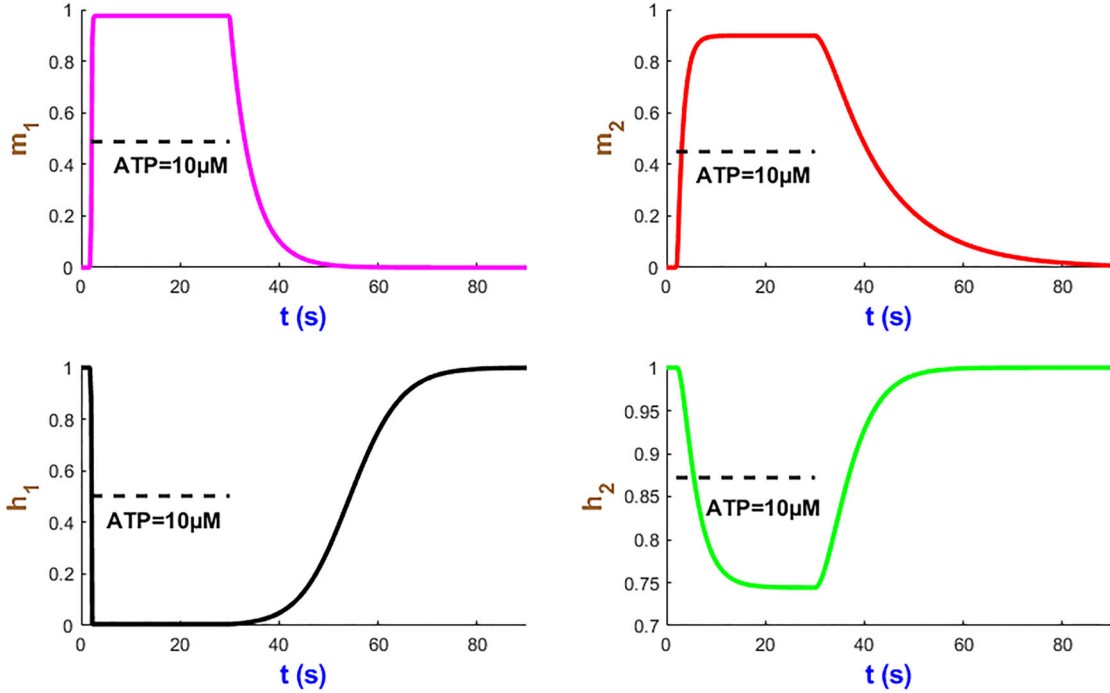

**Figure 17. Temporal dynamics of gating variables in the hP2X$_4$ model at ATP = 10 μM**
The plots relate to the stimulation of the model in Fig. 16.

desensitisation of currents very well, which is a unique kinetic profile for hP2X$_6$Rs (Ormond et al., 2006).

Figure 22*B* and *C* illustrates simulations of the hP2X$_6$ model under two long and short ATP applications. In longer duration of ATP a delayed but sustained

activation phase happens, which is followed by a slow inactivation process. When ATP exposure is short, the current peaks transiently with moderate desensitisation occurring even within the short timescale. In Fig. 22*D*, $m_1$ sharply rises before stabilising. But $m_2$ follows $m_1$ with a

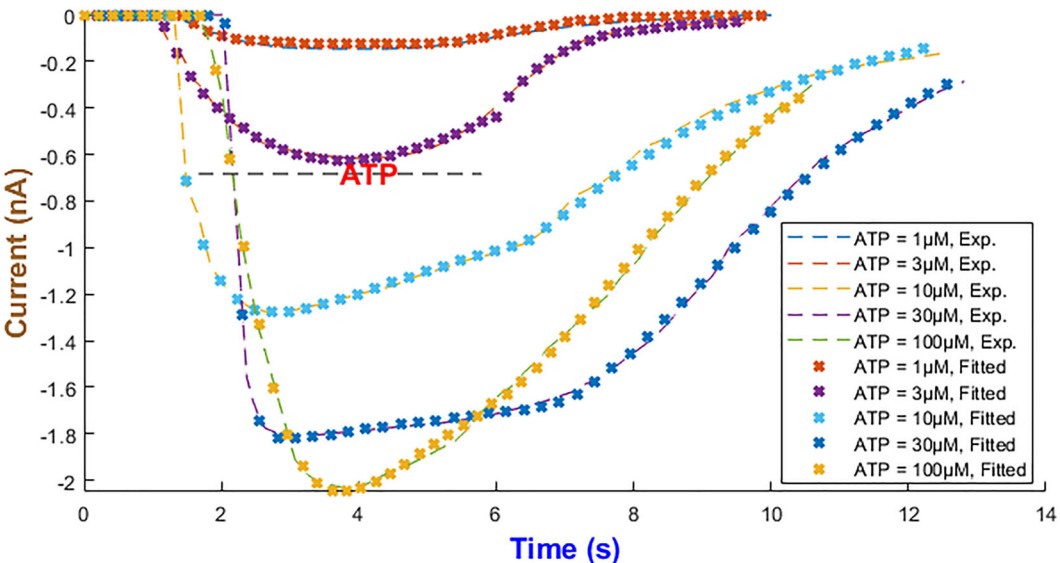

**Figure 18. Validation of hP2X$_5$ model**
The first current form (eqn (6)) can accurately represent the hP2X$_5$R in response to multiple levels of ATP by Bo et al. (2003).

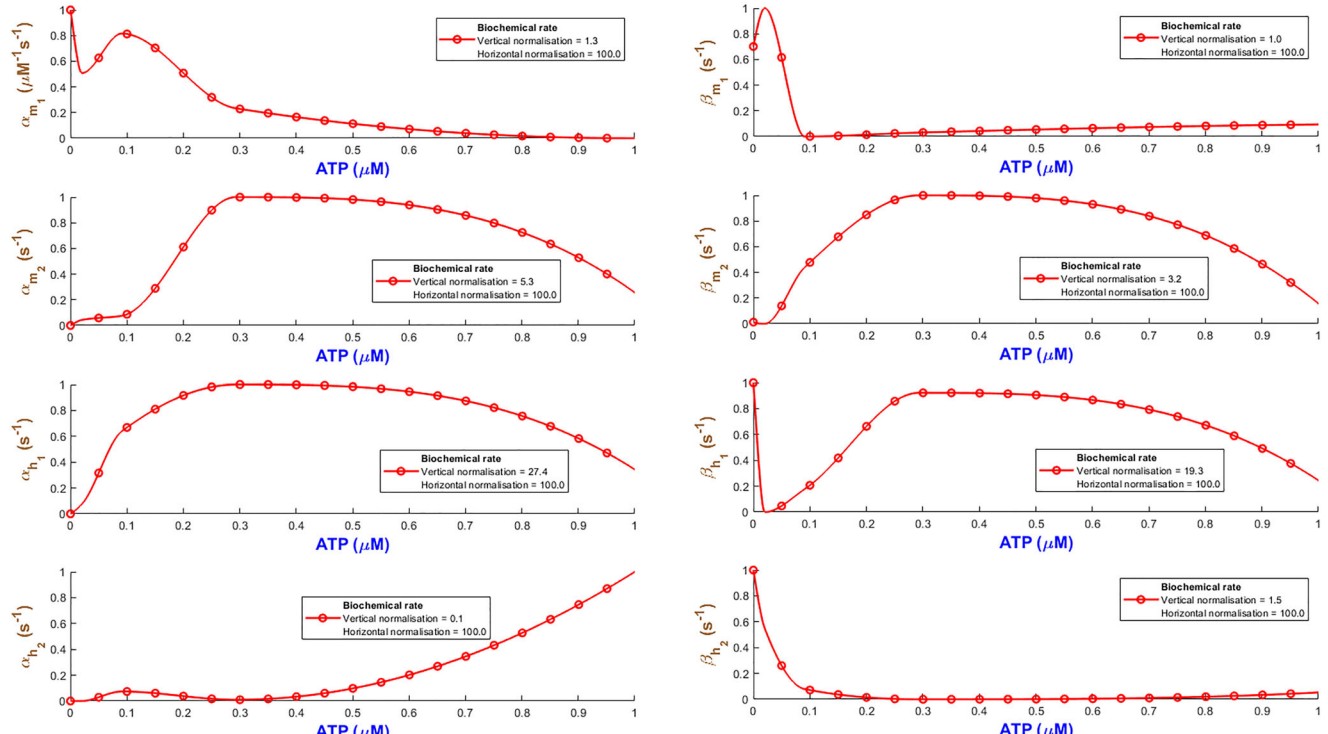

**Figure 19. Fitted and normalised rate constants for the hP2X$_5$ model**
All parameter values were normalised respective to their maximum values (as shown in subplot legends), whereas ATP was normalised respective to 100 μM.

delayed increase. When ATP becomes available, $h_1$ slowly declines and therefore extends the receptor's inactivated state.

## Human P2X₇ model

The hP2X₇ receptor is expressed in immune cells, glia and neurons and has an unusually long C-terminal tail (Gulbransen et al., 2012; Rotondo et al., 2022; Zheng et al., 2024). We identified that $I_{hIR_1}$ effectively characterises the hP2X₇ responses to both nano and micromolar applications of ATP in published datasets (Klapperstück

et al., 2001; Roger et al., 2010). The fitted parameters are listed in Table 1 and Table S1. 12. The fitted model in Fig. 23*A* and *B* shows that the gHH model reproduces slow and fast deactivation in all regimes.

The highly ATP-dependent rate changes appearing in Fig. 24 show that the hP2X₇ receptor takes benefit from a dual regulation mechanism. $\alpha_{m_1}$ decreases continuously (i.e. slower activation at higher levels of ATP). Cooperative binding could have affected the variability in $\alpha_{m_2}$. The strong curvature observable in $\beta_{m_1}$ and $\beta_{m_2}$ can emerge from multiple deactivation states. Two valleys and one peak in $\beta_{m_2}$ indicate that the receptor operates in a very dose-response-dependent manner.

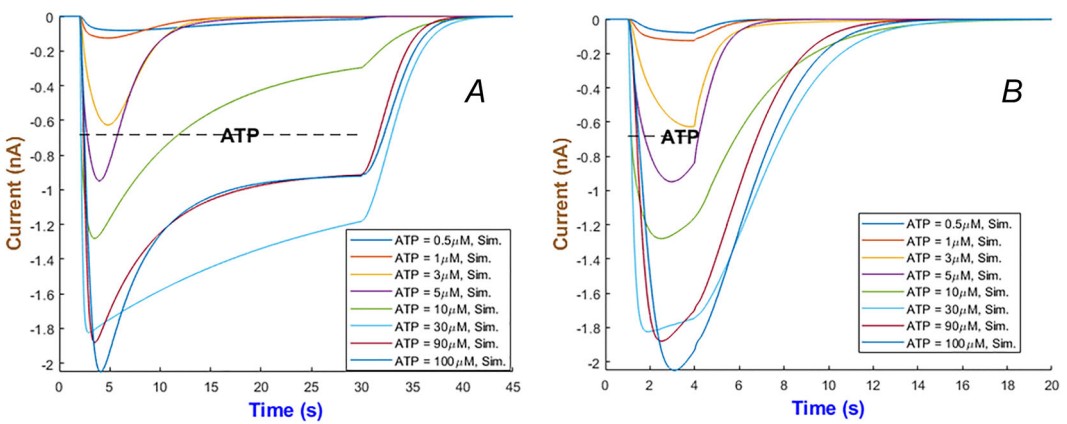

**Figure 20. Simulating the hP2X₅ model**
*A* and *B*, illustrate the dynamic responses of the model over long (28 s) and short (3 s) ATP exposure times, respectively.

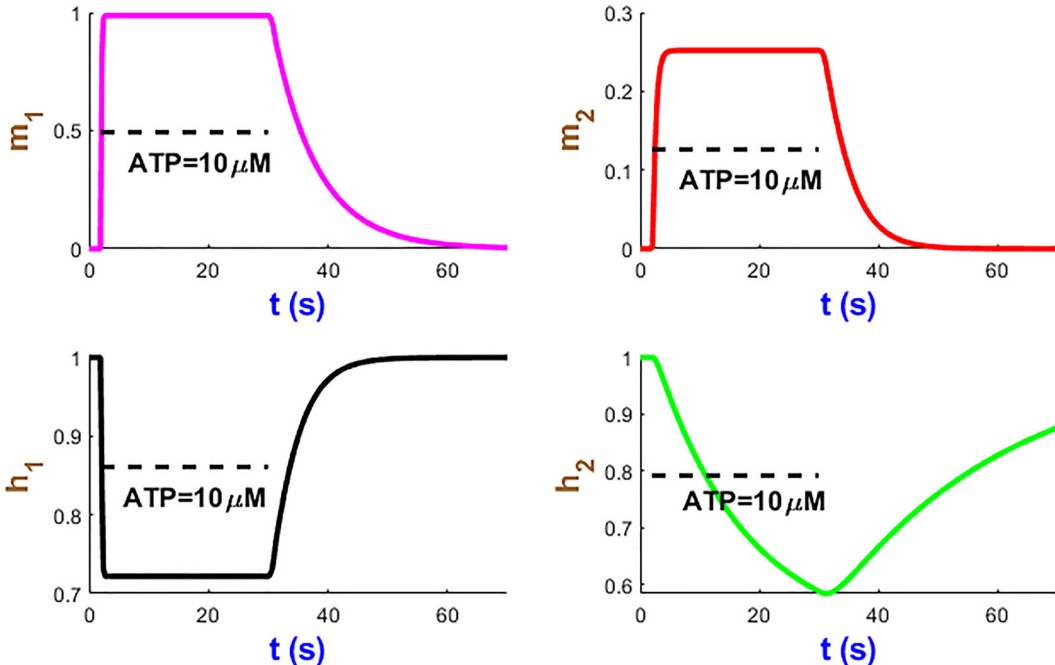

**Figure 21. Predictions of hP2X₅ model's gating variables**
The subplots show how $m_1$, $m_2$, $h_1$ and $h_2$ regulate the differential response rates and recovery patterns.

Inactivation rates are also irregular in accordance with their corresponding activation rates. hP2X$_7$R preserves its function under extended durations of agonist by monotonically increasing $\alpha_{h_2}$. $\alpha_{h_1}$ takes a Gaussian-like path and decreases substantially at higher levels of ATP.

Figure 25$A$–$D$ compares the simulated current responses of hP2X$_7$ under two ATP application protocols. When ATP duration is long, currents plateaus after sustained activation and reach their peaks. This behaviour in which the current remains constant without desensitisation is a defining characteristic of P2X$_7$ (Roger et al., 2008). Note that deactivations in nanomolar levels of ATP for both short and long exposures are slower than those of micromolar ATP.

Under 28 s of 1 mM ATP, the gating variables follow different timelines in Fig. 26. The $m_1$ gate activates quickly and pushes $h_1$ into a biphasic inactivation. In the presence of an agonist, $m_2$ and $h_2$ proceed more slowly. $h_2$ recovers

faster than any other gate and subsequently strongly governs the receptor's return to the resting state. It is worth noting that the slow activation and slow deactivation in $m_2$ is absent in hP2X$_{1-6}$ receptors.

## Human GluA1 model

The hGluA1 receptor mediates millisecond synaptic transmission. It also critically functions in learning and memory. These fast kinetics are strongly affected by glutamate binding and auxiliary protein interactions (Hansen et al., 2021). $I_{hIR_5}$ was the best candidate out of five to fit hGluA1 experimental responses accurately in Coombs et al. (2012) (see Fig. 27) with parameters listed in Tables 1 and S1. 14. Through manual inspection it turns out the functional form of $\varphi(A)$ hints at an exponential dependency on the agonist (namely, $\varphi(A) = e^{-A}$). This function helps the mathematical structure, which is shared

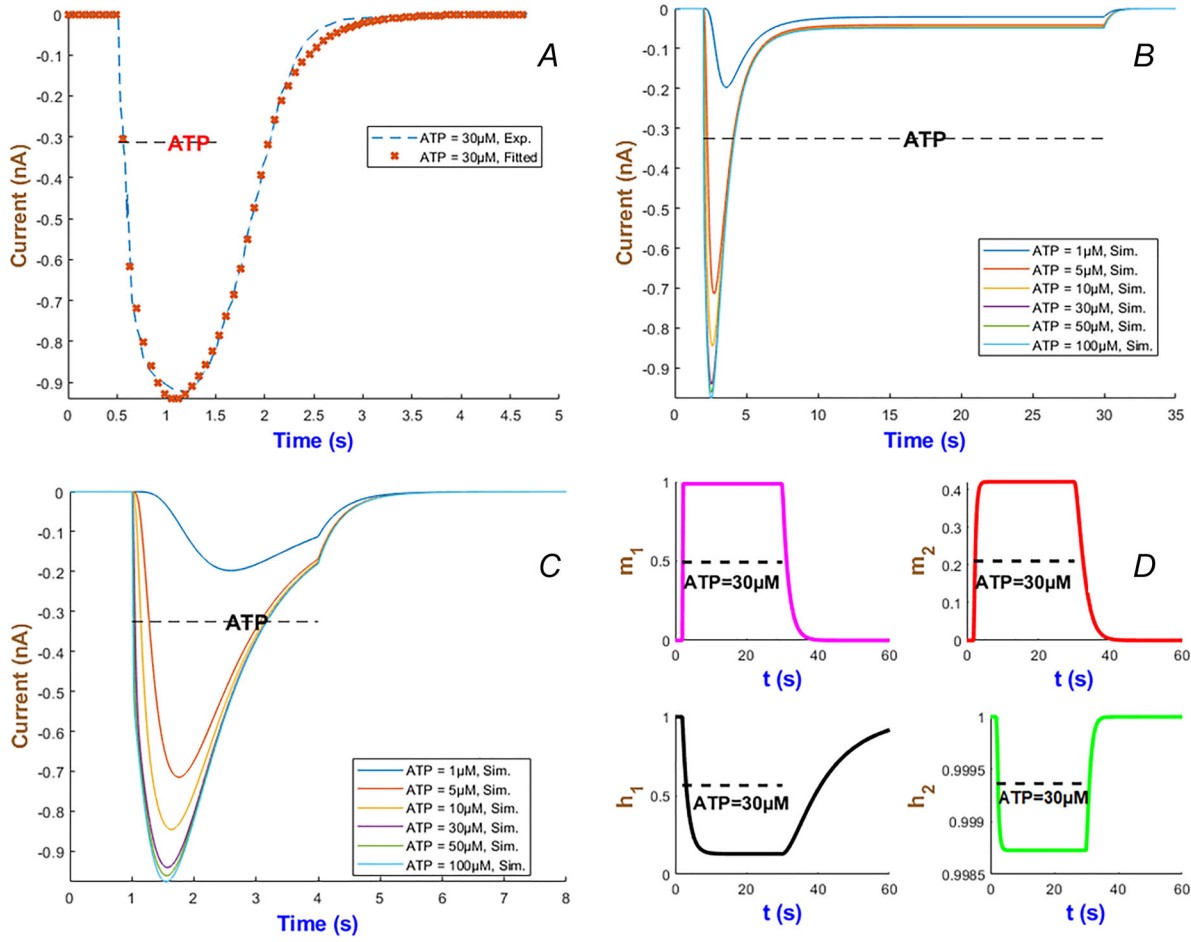

**Figure 22. Fitting and simulation of hP2X$_6$ model responses**
$A$, illustrates fitting the hP2X$_6$ model using the first form of our generalised Hodgkin–Huxley (gHH) model (eqn (6)) in replicating the current dynamics of the hP2X$_6$ receptor at 100 μM ATP documented in (Collo et al., 1996). $A$ and $B$, show the dynamic responses of the hP2X$_6$ receptor model for low and high levels of ATP over long (28 s) and short (3 s) ATP exposure times, respectively, across concentrations ranging from 1 to 100 μM. $D$, shows the time course of the model gating variables for a 28-s application of 30 μM ATP.

with P2X receptors, to be kept invariant. When $A$ is zero, these rates double because $\varphi$ becomes one. Thanks to the negative exponent, $\varphi(A)$ takes a small value in the availability of agonist, and these rates turn out to be independent of $A$. In all simulations $A$ is a time-dependent input (namely $A(t)$) matching the experimental pulse. Any decay from degradation or uptake can be reflected in $A(t)$; here, brief ligand pulses are well approximated as square steps.

The rate constants by which the hGluA1R cooperatively controls high-frequency firing appear in Fig. 28. $\alpha_{m_1}$ and $\beta_{m_2}$ continuously decline, whereas both $\alpha_{m_2}$ and $\beta_{m_1}$ biphasically change. $\beta_{h_1}$ delays inactivation at higher levels of Glu, but $\beta_{h_2}$ speeds it up. The recovery rate $\alpha_{h_1}$ first peaks and then tapers, but $\alpha_{h_2}$ keeps increasing.

Figure 29 characterises how quickly hGluA1 activates and desensitises. In adherence to the high-affinity saturation feature of AMPA receptors, currents stay

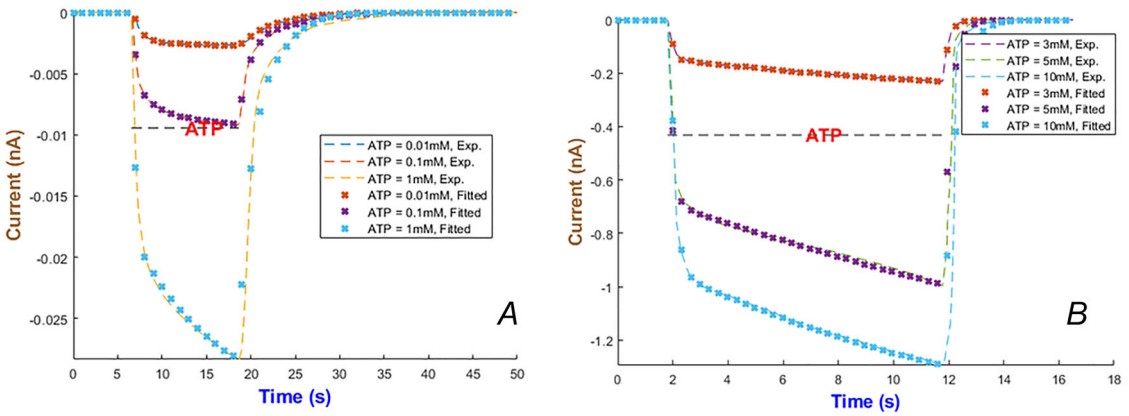

**Figure 23. The hP2X₇ model fitted to experimental currents**
The traces show that the first form of current (eqn (6)) is able to replicate the hP2X₇ currents across low and high levels of ATP concentrations ranging from 0.01 to 1 mM (*A*) and 3 to 10 mM (*B*) published in (Klapperstück et al., 2001; Roger et al., 2010).

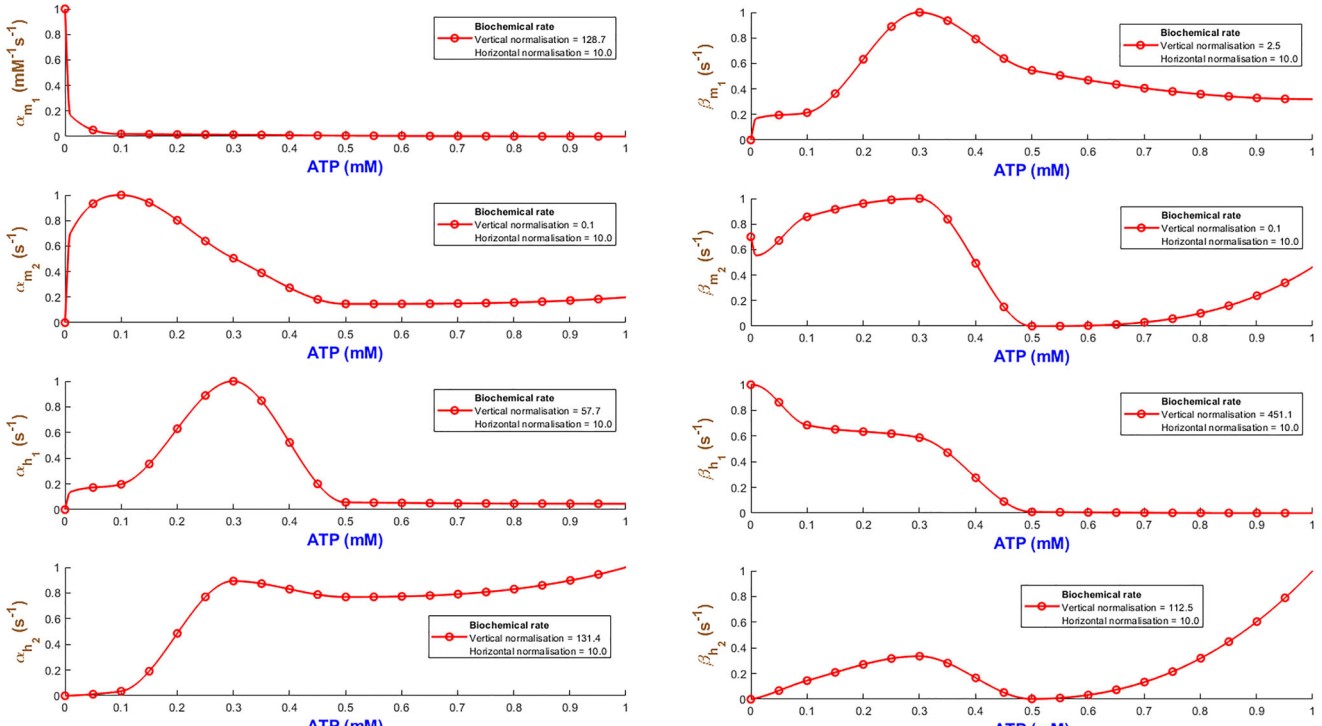

**Figure 24. hP2X₇R's parameters are highly irregular when ATP levels change**
All fitted values are normalised to their maximum rates (see Fig. 23).

active for longer durations before inactivating at higher levels of ligand.

Gating variables of the hGluA1 receptor are shown in Fig. 30. The first three gates exhibit fast biphasic activation and inactivation. Upon glutamate removal $h_2$ returns to its baseline after 30 ms, which agrees with the recovery times of AMPA receptors (Jones & Westbrook, 1996). This machinery contributes to rapid neurotransmission between synaptic events.

### Sensitivity analysis of the model

As discussed throughout this paper, ATP regulates a wide spectrum of P2X-mediated channel properties. To elucidate the model's robustness and provide insight into which parameters mostly influence the output we carried out sensitivity analysis (SA) (Zi, 2011) by varying each rate constant individually (with all others held fixed). Then we measured the resulting change in the simulated whole-cell current (for a specific level of ATP). In this process if the model result does not considerably

change after perturbing parameters, it is said that the sensitivity analysis is robust. We performed sensitivity analysis for all ionotropic currents with respect to 0.1% and 10% perturbations made to rate constants. As seen in Fig. 31 for the hP2X$_4$ model both graphs are similar, which confirms that the optimiser has obtained stable parameters with very low sensitivity to their variations (see Supporting Information S1 for other models). More importantly the whole-cell current exhibits negative and positive sensitivity to some parameters and is mainly affected by $\alpha_{m_2}$, $\alpha_{h_1}$ and $\beta_{h_1}$. Note that in the *y*-axis in Fig. 31 the SA graph shows the proportional change in the whole-cell current when a single parameter is perturbed. Our aim here was simply to demonstrate that the fitted model remains stable when rate constants deviate from their optimal values by amounts comparable to experimental uncertainty. We therefore adopted the classical one-parameter-at-a-time (OPAAT) approach: each rate constant was perturbed by 0.1% to obtain a numerically stable local derivative, and by 10% to bracket the $\approx$5%–15% error typical of ligand-gated channel fits.

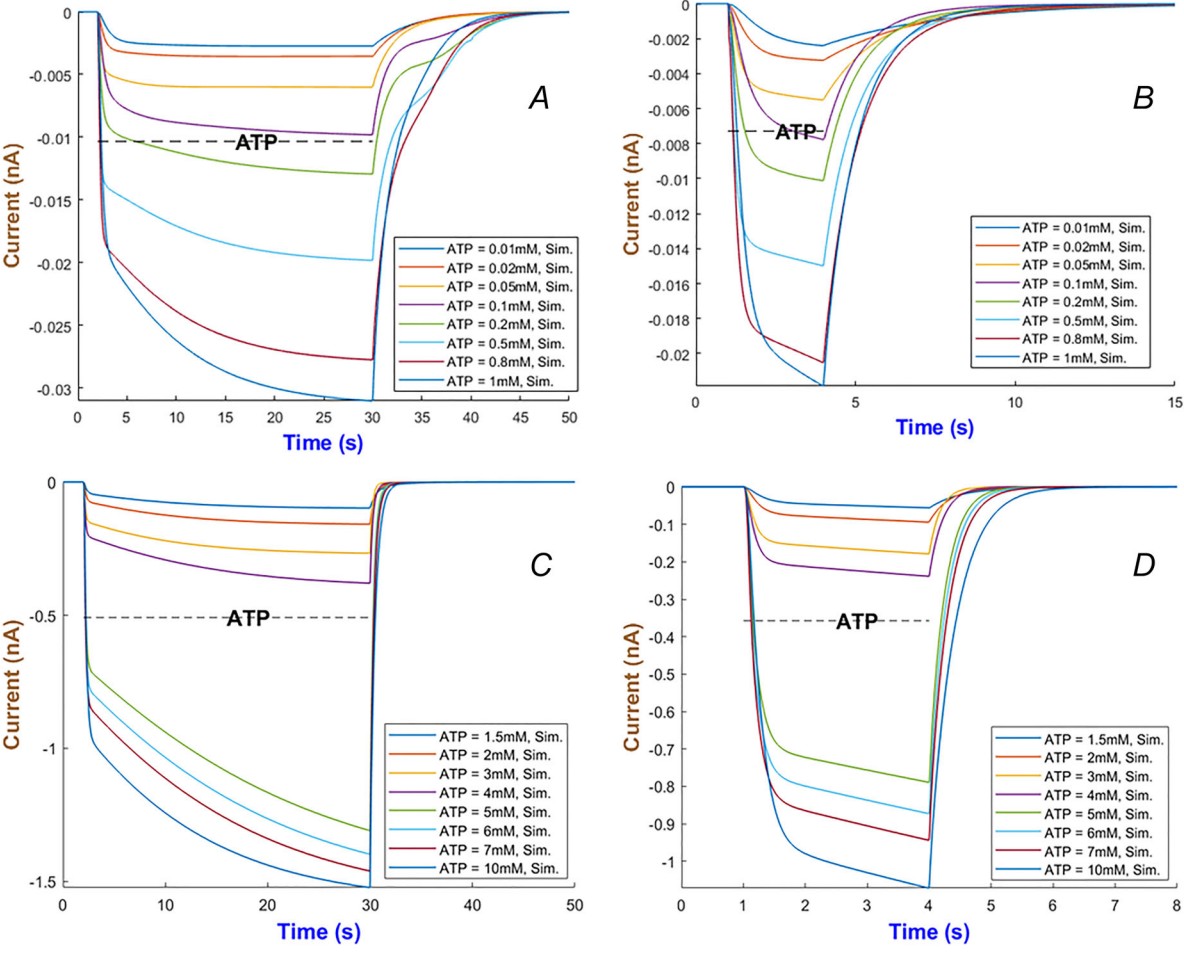

**Figure 25. Model predictions for hP2X$_7$ receptor**
*A–D*, portray the predictions to varying levels and durations of ATP.

The *y*-axis in Fig. 31 reports the relative sensitivity ($\Delta I/I$), so all curves are directly comparable. The near-overlap of the 0.1% and 10% traces confirms that the model output is governed by first-order behaviour and is therefore robust to moderate parameter variation.

### Model summary and cross-receptor comparison

In previous sections the gHH model was studied for different ionotropic receptors. Here, in Table 2, a summary of key features and gating properties for different fits of the unified gHH model to human P2X and AMPA receptors is presented. Details show how each ligand-gated ion channel differs from others and which properties are shared between them in terms of current equation, activation/inactivation profiles along with distinct recovery time scales and distinctive kinetic features.

## Discussion

This study introduces a generalised Hodgkin-Huxley model that successfully captures the diverse gating kinetics of human ionotropic receptors, specifically

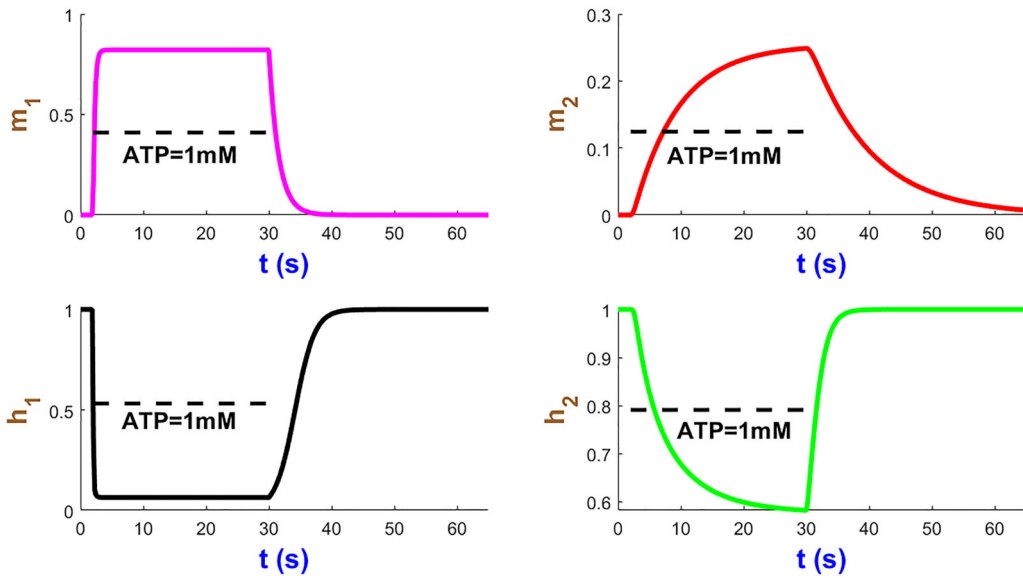

**Figure 26. Time evolution of four hP2X$_7$ gates**
They collectively control the receptor response to a long exposure of ATP.

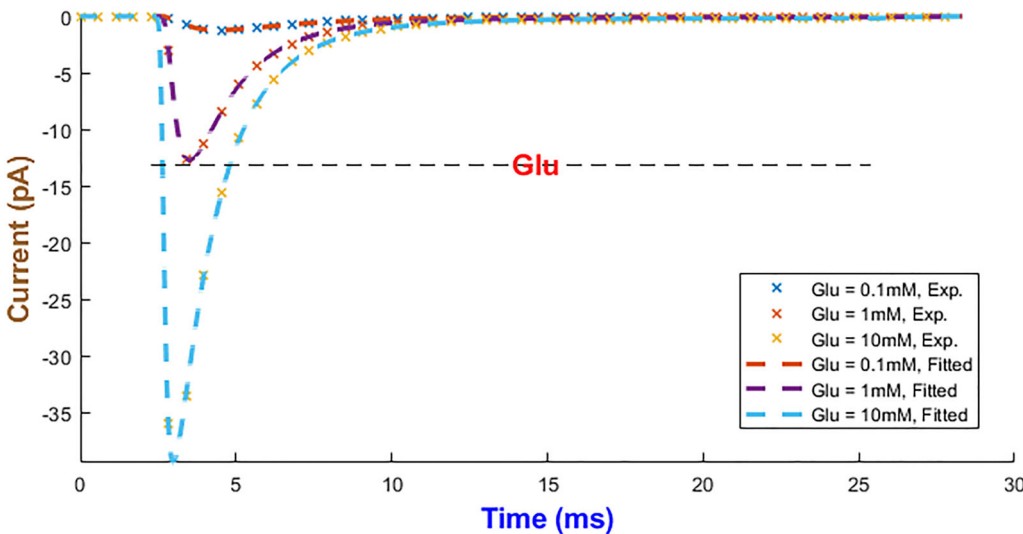

**Figure 27. Fitting results of the hGluA1 model**
Glu concentrations vary from 0.1 to 10 mM. The horizontal axis indicates time in units of milliseconds. Experimental data were extracted from Coombs et al. (2012).

the entire P2X family (hP2X$_{1-7}$) and the AMPA-type glutamate receptor (hGluA1). The classical HH formalism was generalised to ligand-gated receptors by introducing two activation gates and two inactivation gates inter-twined using a law of powers among them. Our broad findings support that the gHH model can characterise a wide family of gating features that are universally common between ion channels – such as activation, inactivation, desensitisation, recovery, binding and cooperativity – across different stimulation protocols. Therefore this work

introduces a unifying alternative to complex multi-state Markov models for ion channel modelling.

Many recent studies illustrate a state-creep phenomenon in Markov-type ion-channel modelling: to capture ever-finer kinetic subtleties, authors successively append layers of closed, open, desensitised or dilated states until the scheme becomes unwieldy. For the P2X$_4$ receptor, Mackay et al. first tried a 22-state network with 58 adjustable parameters; only after expanding to a 34-state, 65-parameter diagram were fast activation

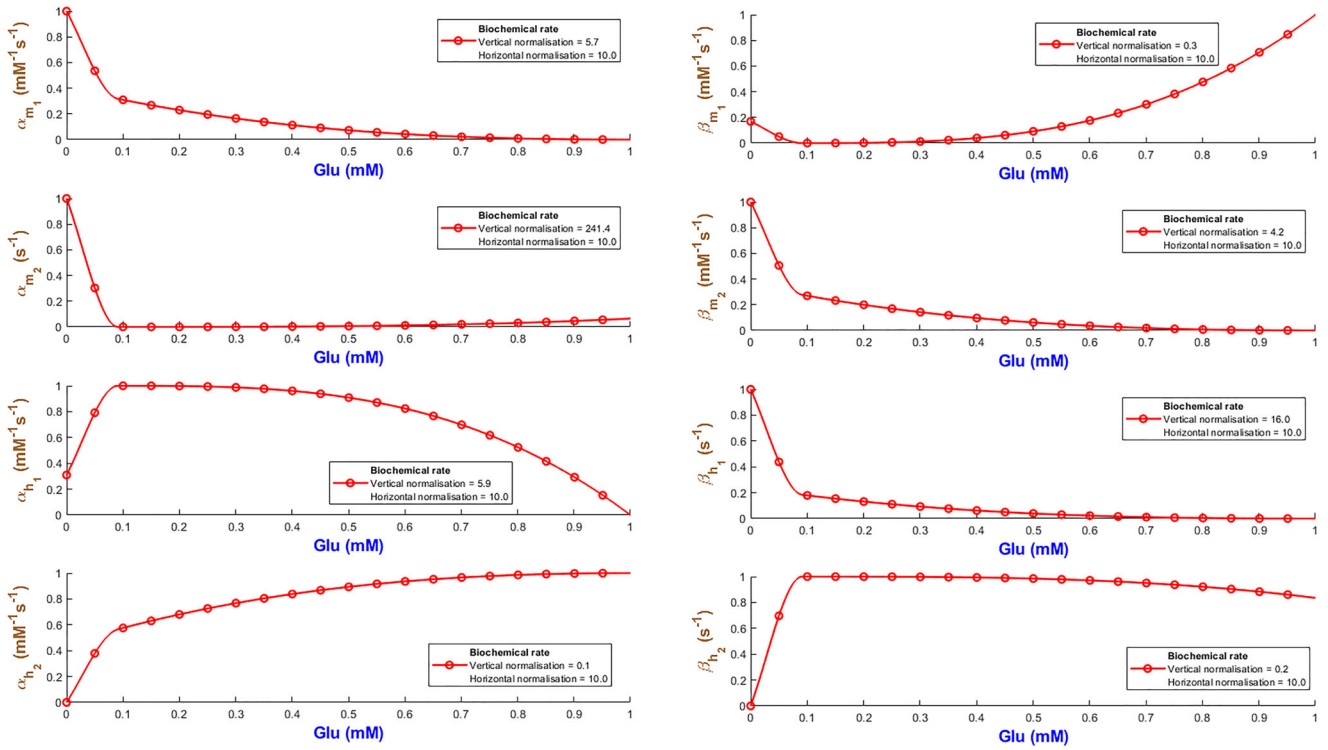

**Figure 28. Fitted biochemical rates for human glutamic receptor**
Note that all quantities are accompanied with their corresponding normalised values.

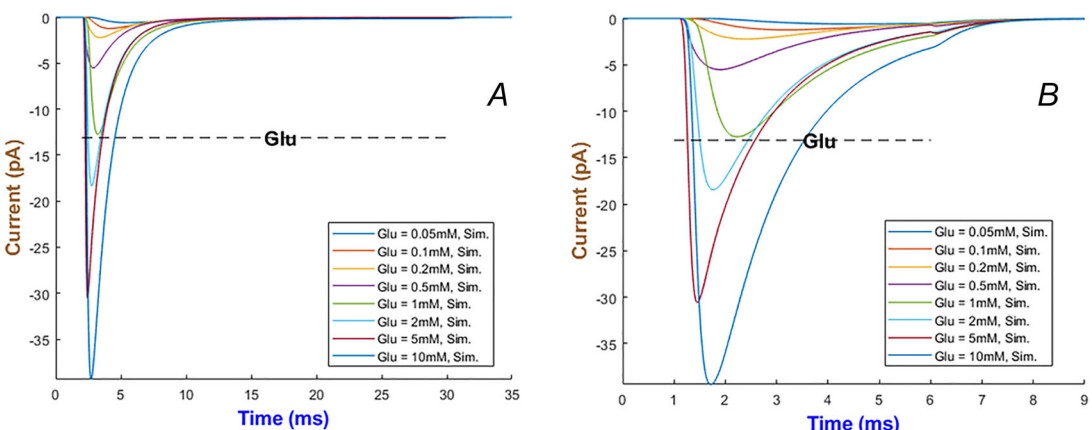

**Figure 29. Simulation of hGluA1 model under a wide range of Glu levels**
Horizontal, dotted bars illustrate the long (28 ms) and short (5 ms) durations of Glu in (*A* and *B*).

and deactivation successfully reproduced (Mackay et al., 2017). A comparable escalation is seen in $P2X_7$ work, where an 8-state naïve/sensitised model grew to 12 states to explain biphasic currents and pore dilatation (Khadra et al., 2013; Yan et al., 2010). Voltage-gated channels follow the same trend: contemporary $Na_v1.5$ descriptions employ 8 to 13 states (Asfaw & Bondarenko, 2019).

Large state spaces pose three recurrent problems, including, identifiability, computational burden and topology choice. Bayesian MCMC analyses have shown that, for large Markov schemes, many different rate sets reproduce the same macroscopic data, leading to practical non-identifiability and divergent mechanistic interpretations (Fink & Noble, 2009; Siekmann et al.,

2011, 2012). Stiff ODE systems with dozens of coupled equations slow down whole-cell or tissue simulations (as well as parameter estimation) to the point where uncertainty analysis becomes prohibitive (Fink & Noble, 2009). Because most receptors lack complete structural or single-channel constraints, deciding which transitions to include is necessarily subjective, and alternative diagrams often fit equally well. Designing a Markov model is itself burdensome: investigators must first hypothesise a state diagram consistent with ligand stoichiometry, then wade through a combinatorial swarm of alternative topologies. Even with automated searches, iterating among candidate networks and constraints remains a labour-intensive task that demands equal parts' biophysical insight and

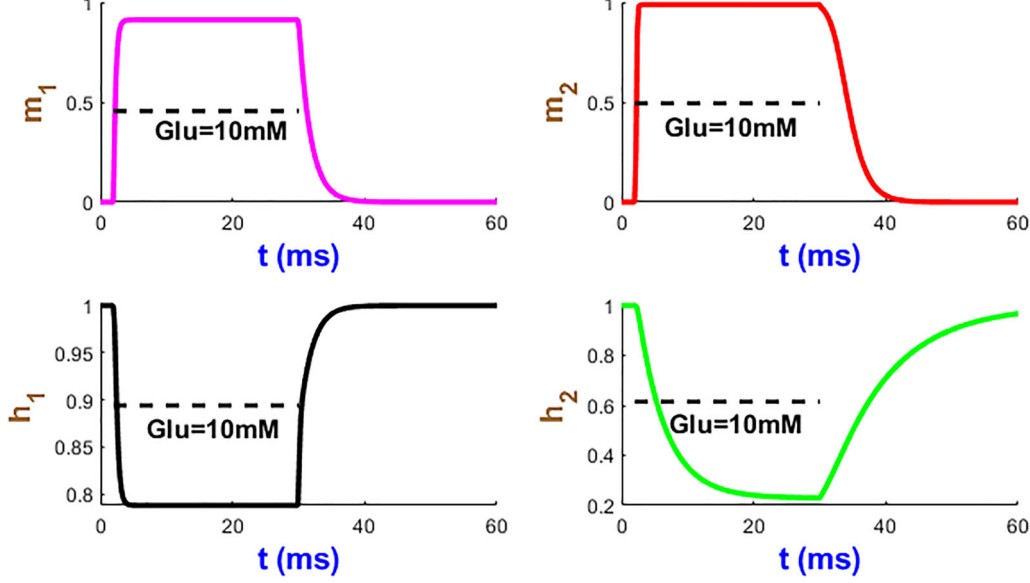

**Figure 30. Channel properties of hGluA1 model**
The interplay of four gates cooperatively shapes the receptor on how to respond to Glu = 10 mM.

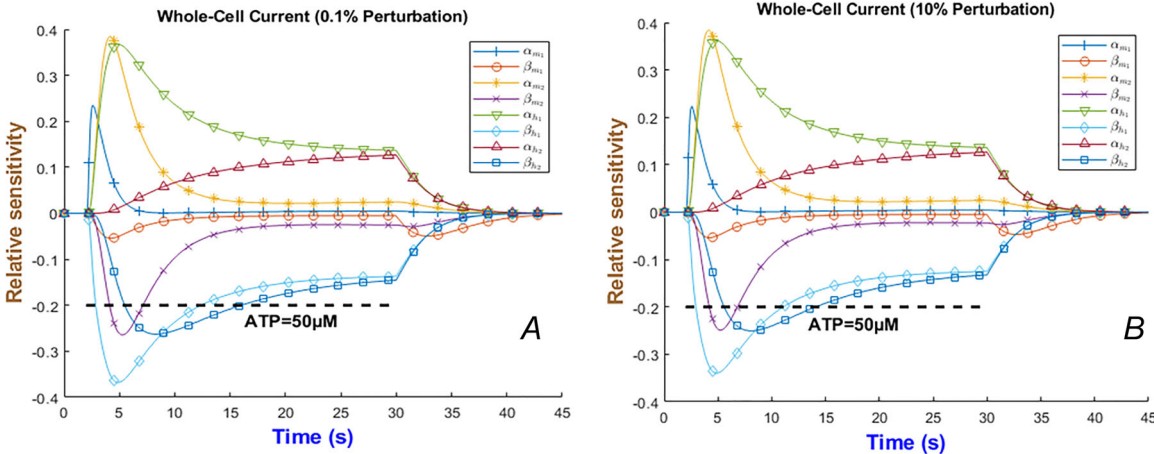

**Figure 31. Sensitivity analysis of the whole-cell current for ATP = 50 μM with respect to the parameter set of hP2X$_4$ receptor**
0.1% (*A*) and 10% (*B*) perturbation of rate constants. The horizontal bar shows the duration of agonist exposure.

**Table 2. Summary of the generalised Hodgkin–Huxley model fitted to $hP2X_{1-7}$ and hGluA1 receptors**

| Receptor | Best current form | Activation behaviour | Desensitisation/plateau | Recovery timescale | Distinctive kinetic feature |
|---|---|---|---|---|---|
| $hP2X_1$ | Equation (7): Multiplicative activation, additive inactivation | Fast; peaks within $< 50$ ms at $\geq 4$ μM ATP | Partial; shallow plateau at high ATP | Seconds | Rapid activation yet sustained residual current at low ATP |
| $hP2X_2$ | Equation (9): Dual-pathway additive-multiplicative | Rapid and sustained activation; biphasic response depending on ATP | Partial inactivation with pronounced dip and recovery at varying ATP; plateau at sustained ATP | Seconds | Dual-pathway gating explains transient 'sag–recovery' profile |
| $hP2X_3$ | Equation (7): Multiplicative activation, additive inactivation | Ultra-fast ($\leq 5$ ms) | Near-complete within $< 100$ ms | Minutes; ATP-dependent (Fig. 13) | Long use-dependent desensitisation typical of nociceptors |
| $hP2X_4$ | Equation (8): Additive activation/inactivation | Moderate (1 s–1.5 s) | Tri-phasic; shallow mid-range plateau | Seconds | Bile-acid-sensitive triphasic inactivation reproduced |
| $hP2X_5$ | Equation (6): Additive activation, multiplicative inactivation | Fast at high ATP; slower at high ATP | Partial; ATP-dependent depth | Seconds | Strong cooperative exponents ($n_{1,2,3} \approx$ 2–3) capture subunit synergy |
| $hP2X_6$ | Equation (6): Additive activation, multiplicative inactivation | Slow onset ($\geq 200$ ms) | Very slow decay | Seconds | Receptor remains open substantially longer, suggesting prolonged signalling roles |
| $hP2X_7$ | Equation (6): Additive activation, multiplicative inactivation | Slow (400 ms, due to Unique delay in $m_2$ gate) then sustained | Minimal; quasi-non-desensitising | Seconds | Plateau current without rundown reproduced over 28 s pulse. Irregular, highly ATP-dependent kinetics |
| hGluA1 | Equation (10): Fully multiplicative gating | Ultra-fast ($\leq 1$ ms) | Rapid desensitisation ($< 5$ ms) | Milliseconds | Exponential $\varphi(A)$ term captures sub-millisecond recovery |

kinetic-modelling expertise. The gHH framework presented herein addresses these issues directly. With only four dynamic variables and a handful of integer exponents it reproduces the kinetics of seven hP2X subtypes and hGluA1 AMPA receptors – behaviours that previously demanded 8- to 34-state schemes. All parameters map onto biologically meaningful quantities (activation, inactivation and recovery rates plus cooperativity indices), which greatly clarifies sensitivity and mutation analyses. Importantly because the number of parameters is limited, these parameters are identifiable with routine data fitting, and their values can be robustly estimated from experiments.

Reducing the model to a few state variables confers substantial computational benefits. Simulations can run sufficiently quickly that Monte-Carlo uncertainty propagation and high-throughput virtual-ligand screening become practicable on standard research hardware, as emphasised in the systematic assessment of speed *versus* complexity by Fink and Noble (2009). At the same time identifiability improves because the compact parameter set can be constrained by conventional voltage- or ligand-jump protocols rather than exhaustive factorial designs. Versatility is retained through interchangeable current expressions that invoke multiplicative, additive or hybrid gating logic. These variants emulate with excellent

accuracy properties like cooperativity, independent branch gating, use-dependence and bi-exponential or sigmoidal responses without redrawing network diagrams. Finely grained Markov models remain invaluable when single-channel or high-resolution structural data justify explicit intermediates such as ligand-binding order, pore-dilatation steps or mutation-specific pathways, but the gHH framework offers a complementary, computationally lean description that preserves mechanistic insight while making large-scale simulations and drug-discovery workflows tractable.

Although the classical HH model is often called phenomenological – it reproduces observed currents without enumerating every microscopic state – its $\alpha/\beta$ rate functions still correspond to physically meaningful transitions; our four-gate generalisation preserves that semi-mechanistic link while remaining far more compact than large Markov models. Therefore the gHH model directly connects electrophysiological recordings to a unified, biophysically interpretable framework for receptor modelling. Even though the gHH equations are written in a forward-dominant form, they can be augmented with reciprocal (h-dependent) backward fluxes (eqns S1.1–S1.2). This feedback adds an explicit return path between activation and inactivation gates – approximating microscopic reversibility – yet it does not enforce the stricter loop constraints of detailed balance. Implementing the variant for hP2X$_4$, hP2X$_7$ and hGluA1 increased the fitting error in every case, indicating that the simpler formulation already captures the macroscopic kinetics; the feedback option remains available for future studies that require a closer link to thermodynamic cycles. Most importantly we showed that there are five possible arrangements within the four model gates by which the generalised current equation can be tailored to receptor-specific responses while the underlying model structure is kept unchanged.

The information gained from the model simulations predicts interesting quantitative/qualitative properties underlying human ionotropic receptors that agree with experiments. This was in part to show that the four simultaneous first-order equations can describe ion channels that operate over multiple timescales – *viz.* millisecond/second activation to prolonged recovery across several minutes. Ionotropic receptors are important ion channels in neurophysiology, pharmacology and computational neuroscience; therefore this work accounts for a fundamental move towards a unified theory of single-channel and/or whole-cell electrophysiological modelling. Our gHH formalism takes computational efficiency into consideration while retaining mechanistic intuition into receptor functions. In contrast the high dimensionality of Markov models gives rise to critical issues in state space and parameter identifiability and estimability. These introduce several barriers to using them in scalable simulations or basic translational research. There is a recognised need in computational neuroscience for models that can accurately represent the intricate behaviours of ion channels and receptors while remaining computationally efficient, interpretable and adequately simple to capture necessary kinetic details (Cannon & D'Alessandro, 2006; Herz et al., 2006; Huys et al., 2006; Podlaski et al., 2017). Our work is an attempt to bring molecular details into systems-level phenomena (De Schutter, 2008) and biological variability and plasticity in neural systems (Marder, 2011) through robust principles. By balancing mechanistic depth with scalability, our mathematical model resolves the long-standing trade-off between complexity and interpretability by laying the foundation for a unified mathematical approach to ligand-gated ion channels and beyond.

This framework can be reused to construct scalable cellular models which investigate P2X-mediated Ca$^{2+}$ signalling in immune and non-neuronal cells. A growing wealth of evidence shows that when ion channels begin to malfunction, neuroinflammation and neurodegeneration occur (Ai et al., 2023; Burnstock & Boeynaems, 2014; Chen et al., 2022; Luo et al., 2021; Poshtkohi et al., 2024; Sandoval et al., 2024). Combining purinergic human models is instrumental in studying how neuron-glia interactions contribute to synaptic and glial transmission in the brain (Linne et al., 2022; Liu et al., 2021; Thomasi & Gulbransen, 2023). The gHH model can simulate scenarios under which either ATP or glutamate varies in time, thanks to the explicit dependence of its rate constants on agonist concentration. Therefore our framework enables building complex models that simulate the interaction of ligand-gated receptors under dynamic ATP levels with voltage-gated ion channels during repetitive firing of action potentials in synaptic transmission through neurons and the cardiac cycle through cardiac myocytes (Burnstock, 2017; Zimmermann, 2016).

Although individual P2X receptor subtypes and certain heteromeric forms (e.g. P2X$_{2/3}$) are well studied, investigating the simultaneous activation of multiple distinct subtypes remains a challenge experimentally due to receptor redundancy, overlapping conductances and difficulty in isolating subtype-specific currents (Khakh, 2006; North, 2002). For example the dissection of P2X$_4$ and P2X$_7$ currents in microglial cells is difficult as both receptors independently respond to ATP. This specifically complicates the interpretation of their combined effects (Trang et al., 2020). On the contrary experimental characterisation of ion channels using isolated conditions or heterologous expression systems often fails to replicate their native environment (Burnstock & Boeynaems, 2014). Our computational framework can facilitate the simultaneous activation of multiple P2X receptors in the *in silico* exploration of realistic physiological interactions

and ionic currents. This approach is especially relevant in microglia, astrocytes, macrophages, enteric glia and myenteric neurons, where multiple P2X subtypes are involved in immune responses, neuroinflammation and synaptic signalling (Illes et al., 2020). Thus our model offers a valuable quantitative tool that can complement experimental findings and generate testable predictions. It can also be used as an alternative to expensive and technically challenging *in vitro* and *in vivo* studies (Yan et al., 2010). Such an in-silico model can serve as a hypothesis-generating tool in addition to being a platform for screening potential drugs.

As flexibility is a key feature of our framework, we plan to extend the model beyond P2X and AMPA receptors. The equations can be adjusted to represent the cooperative nature of voltage-gated $Na^+$, $K^+$ and $Ca^{2+}$ channels (Huang et al., 2024) in which the term $A$ is removed and voltage dependencies are instead absorbed by the model rate constants. Every rate constant in the model can be modelled as a bivariate function of two ligands, while a multiplication over them can replace the term $A$. Therefore, certain multi-ligand channels – such as $IP_3$ receptor ($IP_3R$) that responds to both calcium and $IP_3$ (Smith et al., 2023) – can sit on top of the gHH formalism. Finally the generalised model should be able to be straightforwardly applied to such ion channels as NMDA, GABA, glycine, nACh or TRP (Dupuis et al., 2023; Ghit et al., 2021; Rather et al., 2023) that bind to a single triggering agent.

One limitation of our framework is that finding the proper current form and model exponents is presently done manually, which is time-consuming. Concretely we fit each of the five candidate current forms to the data and keep the one that yields the lowest summed-squared error. This selection step can be automated by systematically evaluating SSE across all candidate forms in a parallel optimisation workflow to speed up determining the best mathematical structure for the receptor(s) in question. This is beneficial to researchers who want to build receptor models for different species. In addition studying the genetic mutations of a specific receptor and comparing its channel properties with its equivalent reference model will become feasible by this extension.

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

## Additional information

### Data availability statement

All digitised datasets, model code, and parameter files that support the findings of this study are openly available at: https://github.com/poshtkohi/gHH. These materials are provided under an open licence and enable full reproduction of the results reported here.

### Competing interests

No competing interests declared.

### Author contributions

All authors have approved the final version of the manuscript and agree to be accountable for all aspects of the work. All persons designated as authors qualify for authorship, and all those who qualify for authorship are listed.

### Funding

No funding.

### Acknowledgements

Alireza Poshtkohi gratefully acknowledges the foundational contributions of Alan L. Hodgkin and Andrew F. Huxley, whose quantitative work on the ionic mechanisms underlying the action potential set the standard for data-driven modelling in electrophysiology and directly inspired this study to generalise Hodgkin-Huxley model developed here. Alireza Poshtkohi also acknowledges Michael Faraday's experimental researches in electromagnetism and electrochemistry; his empirical, data-first method directly inspired the data-driven modelling strategy used in this work. Any remaining errors are Alireza Poshtkohi's own.

## Keywords

electrophysiological modelling, Hodgkin–Huxley model, human glutamatergic receptors, human ionotropic receptors, mathematical modelling, P2X receptors

## Supporting information

Additional supporting information can be found online in the Supporting Information section at the end of the HTML view of the article. Supporting information files available:

**Peer Review History**
**Supporting Information**

