## [Peer Review History · The Journal of Physiology]

Generalised Hodgkin-Huxley Model Captures Human P2X and AMPA Receptor Currents

Alireza Poshtkohi and Brian David Gulbransen
DOI: 10.1113/JP288880

Corresponding author(s): Alireza Poshtkohi (a.poshtkohi@herts.ac.uk)

The following individual(s) involved in review of this submission have agreed to reveal their identity: Robert H Cudmore (Referee #2)

Review Timeline:

Submission Date:	13-Mar-2025
Editorial Decision:	04-Jun-2025
Revision Received:	03-Jul-2025
Accepted:	03-Oct-2025

Senior Editor: Kim Barrett

Reviewing Editor: Brian Delisle

Transaction Report:

Dear Dr Poshtkahi,

Re: JP-RP-2025-288880 "**Generalised Hodgkin-Huxley Model Captures Human P2X and AMPA Receptor Currents**" by Alireza Poshtkahi and Brian David Gulbransen

Thank you for submitting your manuscript to The Journal of Physiology. It has been assessed by a Reviewing Editor and by 2 expert referees and we are pleased to tell you that it is acceptable for publication following satisfactory revision.

REVISION CHECKLIST:

We look forward to receiving your revised submission.

Yours sincerely,

Kim Barrett
Senior Editor
The Journal of Physiology

EDITOR COMMENTS

Reviewing Editor:

Comments to the Author:

Both referees agree that this work is likely to significantly impact the field. However, there are several issues that should be addressed. This includes issues regarding resource sharing as identified by referee 1 and further discussion on the approach used as identified by referee 2. Each referee also identified minor concerns that, if addressed, would improve the overall quality of the work.

REFEREE COMMENTS

Referee #1:

The manuscript "Generalized Hodgkin-Huxley Model Captures Human P2X and AMPA Receptor Currents" develops a modified Hodgkin and Huxley framework to model ionotropic receptors. A key benefit in the model design is 5 different sets of equations that are designed to capture receptor-specific cooperativity, binding kinetics, and desensitization. The model is then used to fit several P2X (7 subtypes) and mGluA1 receptors. In all cases, human receptor data was fit, thus increasing the importance of the results.

The development of the model is excellent, as well as the description in the methods of each piece of all the equations with an intuitive description of how to interpret them. Once a receptor subtype is fit, the analysis of activation and inactivation kinetics (as well as other model equations) is accompanied by a very intuitive description of the function of the model.

For example, in the results of Figure 3, "Such detailed mapping for rate constants across a wide range of ATP concentrations lays a foundation for understanding how each receptor plays its role in physiological conditions and what happens when the receptor is dysregulated in pathological states". I could not agree more! The formalism of this new model will surely lead to novel discoveries in the detailed biophysics of pathological states and with this has the potential to lead to targeted drug discoveries.

To summarize, the new and novel model is sure to impact research on ionotropic receptors in health and disease, will provide excellent insight into physiological mechanisms, and is highly original.

I do have one major concern on data sharing, otherwise, I only have minor concerns below.

Major Concerns (Data sharing)

1) One major concern is with providing enough information and sharing enough data that others could extend this work. This would surely benefit the scientific community and be more in line with current data sharing requirements in both Europe and at the NIH.

1.1) Can the authors provide the source code for the implementation of the model equations? Likewise, can the authors provide the source code used to fit the model? In the former case, the implementation of the model is not mentioned aside from equations, and in the latter case it is briefly outlined as being done in Matlab. This could all be shared in, for example, a GitHub repository.

1.2) Throughout the manuscript, the model is fit to experimental data taken from a range of experimental manuscripts (8 or so, one for each of P2X and AMPA). What is not clear is how the authors extracted the raw current traces from those manuscripts (to perform the fit). Were the raw traces provided by the source experimental manuscripts as raw data files? Or did the authors somehow extract the raw data (current) from the published figures? It would be good to see a summary of how this was done (even if just in the supplemental). Finally, it would be good to provide raw data files with the currents extracted from the source manuscripts that were then fit.

1.3) The supplemental data files need to be better described and organized. I opened a few of them and they seem to be text (.dat) files with two columns of numbers. It would be nice to know, (i) what does each file correspond to, there are many .dat files and I do not know what they refer to, and (ii) what are the units on the corresponding two columns of numbers in each .dat file? Without providing a better description of the organization and units in all the .dat files, they are not that useful.

Minor Concerns

1) There is a huge amount of model fitting (which in itself is great) and each fit is accompanied by an excellent description which gives both modelers and importantly non-modeler an intuitive understanding of how the model is operating. Yet, this is all buried in the results (which is fine). It might be nice to have a table summarizing the intuitive descriptions for how the model is working for all the P2X and AMPA fits? Maybe a table highlighting key points for each receptor, or key differences in the model between different types of receptors.

2) In future work it would be interesting to see a model run where ATP concentration is modulated in real time. The current model fits parameters and then calculates the current for one concentration of ATP. It would be great to see a model run for dynamic levels of ATP as can occur during action potential firing (in neurons and cardiac myocytes for example).

3) Fig 1b. What are the solid and dotted lines in Fig1b? Do they somehow encapsulate "The model allows for precisely replicating a wide spectrum of receptor dynamics consisting of activation, inactivation, desensitization and recovery in different physiological condition".

4) In Table 1, how were hGluA1 parameters estimated? In the results, the authors are fitting data from a manuscript on hP2X subtypes? Does this data also contain hGluA1 currents?

5) In general, the authors did an excellent job of giving a narrative explanation of how the equations work. One minor question, in the

first order equation 1-4, what are the units on agonist concentration "A"? Does it matter, I am not sure? Is it molar?

6) Fig5. The figure is good as it is but want to suggest merging m1/m2 into one plot and h1/h2 into another? That could be done for all m1/m2 plots throughout the manuscript? It is up to the authors.

7) Fig13 shows recovery time for hP2X₃ receptor. Is this recovery unique to hP2X₃ versus 1/2? This figure shows recovery to a brief pulse of ATP (2 s), what about a longer application? Does that alter recovery?

8) The sensitivity analysis to the P2X models is an excellent plus. Yet, a 0.1 or 1 percent tweak in the parameters seem small but this may be acceptable given the particular sensitivity analysis performed. If this is the case, it should be mentioned.

9) In Fig30, the authors should explain what "Relative Sensitivity" is. Is it the change in the parameter before/after changing model fits by 0.1 and 1 %? Basically, I am not sure what the Y-axis represents?

Referee #2 (see attachment):

This study proposes a generalized version of the Hodgkin-Huxley model that incorporates ligand-gated receptors, rather than being limited to voltage-gated channels. It aims to provide a unified, mechanistic mathematical framework for understanding the gating properties of P2X receptors and glutamate receptors. The model extends the traditional Hodgkin-Huxley formulation by including two activation gates, two inactivation gates, and coefficients that capture cooperative ligand binding. This novel approach is anticipated to be broadly applicable to physiological research and drug discovery.

The authors present a thorough evaluation of model fits for several ATP-dependent P2X channel isoforms across varying ATP concentrations, as well as for a glutamate receptor channel. While the results highlight the versatility of the proposed modeling framework, several important considerations merit discussion. These include the model evaluation strategy and the trade-offs involved in using this approach compared to established Markovian models. Additionally, there is some concern that the fitted models, in their current form, may not yield sufficiently novel insights to engage a broad Journal of Physiology readership.

Major:

- A robust discussion of advantages and disadvantages of existing Markov models, like from Mackay et al, would be helpful in placing the novelty of this method in context (<https://journals.plos.org/ploscompbiol/article?id=10.1371/journal.pcbi.1005643>).
- Hodgkin-Huxley has been a useful phenomenological model for describing ion channel activity, and as the author indicates, its simplicity limits its interpretability. The authors' extended model adds gating terms, but still seems to be lacking in physical factors such as reversibility and detailed balance (such as in Figure 1B) that arises naturally in Markovian models. Would these omissions limit the interpretability of this extended model?

Minor:

- Line 691 Could the sensitivity analyses be explained in greater detail? In this elaboration, discussion justifying the advantage of their approach relative to more conventional ones (e.g. a figure of merit is evaluated across a parameter range) would be appreciated. The sensitivity analyses are based on comparisons of predicted currents for perturbed parameters relative to those predicted with best-fit parameters. The parameter perturbations (0.1 and 1% respectively) seem very small, relative to the experimental error one might encounter with determining those parameters.
- Line 254 indicates that only one current form is 'appropriate' for each channel. How was this determined exactly? Also, do they try models with fewer or more gates to assess the quality of fitting?
- Line 311 indicates that optimal beta_{m2} values vary non-monotonically with ATP concentration. Could this be evidence of

there being different states, such as low affinity and high affinity states that wouldn't necessarily be captured by assuming one state for m2? In Line 397, this is alluded to in the suggestion that there may be multiple energetic barriers.

- Line 133. are there any constraints used to determine kinetic parameters when the system is in steady state? In other words, are any measures taken to hold some states fixed, such that a smaller number of parameters can be optimized for a given observable? As it stands, Eqn 9 has at least four free parameters to fit to a 1D line, therefore it is very likely that the parameter solutions are not unique.
- Line 133. How (or why) was the modulation factor of $p(A)=\exp(-A)$ chosen? It seems like the primary effect would be to maintain a small or potentially negative value for states like m1 at low substrate concentrations.
- Line 134 Is there a physical basis for determining or interpreting the values of n_i ? Clearly they add nonlinearity to the rate equations, akin to cooperativity, but how cooperativity arises from the gates is unclear. For instance, how does $n_4=5$ for hP2X4 channels that are trimeric in nature?
- Line 282 - are the values of n constrained in the optimization • strategy?
- Line 318 However, αh_2 shows a distinct sensitivity <-- typo
- Considerable heterogeneity in receptor expression is anticipated, how would the strategy reflect this in its parameter optimization?
- Ln 226 How is goodness of fit determined and used in this study? I believe this is implied in the Model fitting section (i.e. line 245).
- Ln 499 The authors indicate that the recovery time for hP2X3 was 'confirmed by human experimental studies'. Could this be briefly elaborated in the text, such as the cell type, whether they used primary cells, etc.
- Line 662 indicates that the functional form of $\phi(A)$ is exponential in A. Have the authors considered the possibility that A may also be time dependent, given that both nucleotides and glutamate can be degraded?
- Line 715 argues that their unified framework could obviate the fitting of 'complex, multi-state Markov models'. In my opinion, it would probably be helpful to discuss the advantages and limitations of both approaches. As I see it, the model proposed here could be more phenomenological in nature, but cheaper to parameterize, whereas a Markovian model may be more mechanistic, albeit have more free parameters to determine.
- Final paragraph beginning with Line 769 - the 'manual' strategy of determining the current form was not specified. I would advise at least discussing numerical strategies available to select between competing models. The line regarding supercomputers and parallel optimization is vague and extrapolative.

END OF COMMENTS

Poshtkohi J Physiology Review JP-RP-2025-288880

This study proposes a generalized version of the Hodgkin-Huxley model that incorporates ligand-gated receptors, rather than being limited to voltage-gated channels. It aims to provide a unified, mechanistic mathematical framework for understanding the gating properties of P2X receptors and glutamate receptors. The model extends the traditional Hodgkin-Huxley formulation by including two activation gates, two inactivation gates, and coefficients that capture cooperative ligand binding. This novel approach is anticipated to be broadly applicable to physiological research and drug discovery.

The authors present a thorough evaluation of model fits for several ATP-dependent P2X channel isoforms across varying ATP concentrations, as well as for a glutamate receptor channel. While the results highlight the versatility of the proposed modeling framework, several important considerations merit discussion. These include the model evaluation strategy and the trade-offs involved in using this approach compared to established Markovian models. Additionally, there is some concern that the fitted models, in their current form, may not yield sufficiently novel insights to engage a broad *Journal of Physiology* readership.

Major:

- A robust discussion of advantages and disadvantages of existing Markov models, like from Mackay et al, would be helpful in placing the novelty of this method in context (<https://journals.plos.org/ploscompbiol/article?id=10.1371/journal.pcbi.1005643>).
- Hodgkin-Huxley has been a useful phenomenological model for describing ion channel activity, and as the author indicates, its simplicity limits its interpretability. The authors' extended model adds gating terms, but still seems to be lacking in physical factors such as reversibility and detailed balance (such as in Figure 1B) that arises naturally in Markovian models. Would these omissions limit the interpretability of this extended model?

- Line 691 Could the sensitivity analyses be explained in greater detail? In this elaboration, discussion justifying the advantage of their approach relative to more conventional ones (e.g. a figure of merit is evaluated across a parameter range) would be appreciated. The sensitivity analyses are based on comparisons of predicted currents for perturbed parameters relative to those predicted with best-fit parameters. The parameter perturbations (0.1 and 1% respectively) seem very small, relative to the experimental error one might encounter with determining those parameters.

Minor

- Line 254 indicates that only one current form is 'appropriate' for each channel. How was this determined exactly? Also, do they try models with fewer or more gates to assess the quality of fitting?
- Line 311 indicates that optimal β_{m2} values vary non-monotonically with ATP concentration. Could this be evidence of there being different states, such as low affinity and high affinity states that wouldn't necessarily be captured by assuming one state for $m2$? In Line 397, this is alluded to in the suggestion that there may be multiple energetic barriers.
- Line 133. are there any constraints used to determine kinetic parameters when the system is in steady state? In other words, are any measures taken to hold some states fixed, such that a smaller number of parameters can be optimized for a given observable? As it stands, Eqn 9 has at least four free parameters to fit to a 1D line, therefore it is very likely that the parameter solutions are not unique.
- Line 133. How (or why) was the modulation factor of $p(A)=\exp(-A)$ chosen? It seems like the primary effect would be to maintain a small or potentially negative value for states like $m1$ at low substrate concentrations.
- Line 134 Is there a physical basis for determining or interpreting the values of n_i ? Clearly they add nonlinearity to the rate equations, akin to cooperativity, but how cooperativity arises from the gates is unclear. For instance, how does $n4=5$ for hP2X4 channels that are trimeric in nature?
- Line 282 - are the values of n constrained in the optimization strategy?
- Line 318 However, $\alpha h2$ shows a distinct sensitivity <-- typo
- Considerable heterogeneity in receptor expression is anticipated, how would the strategy reflect this in its parameter optimization?
- Ln 226 How is goodness of fit determined and used in this study? I believe this is implied in the Model fitting section (i.e. line 245).
- Ln 499 The authors indicate that the recovery time for hP2X3 was 'confirmed by human experimental studies'. Could this be briefly elaborated in the text, such as the cell type, whether they used primary cells, etc.

- Line 662 indicates that the functional form of $\phi(A)$ is exponential in A . Have the authors considered the possibility that A may also be time dependent, given that both nucleotides and glutamate can be degraded?
- Line 715 argues that their unified framework could obviate the fitting of 'complex, multi-state Markov models'. In my opinion, it would probably be helpful to discuss the advantages and limitations of both approaches. As I see it, the model proposed here could be more phenomenological in nature, but cheaper to parameterize, whereas a Markovian model may be more mechanistic, albeit have more free parameters to determine.
- Final paragraph beginning with Line 769 - the 'manual' strategy of determining the current form was not specified. I would advise at least discussing numerical strategies available to select between competing models. The line regarding supercomputers and parallel optimization is vague and extrapolative.

Generalised Hodgkin-Huxley Model Captures Human P2X and AMPA Receptor Currents

Alireza Poshtkahi and Brian D. Gulbransen

The authors wish to thank the respected referees and the Journal of Physiology editorial team for the time and effort in evaluating our paper. Their insightful comments on the paper led us to an improvement of the work. Our revisions reflect all referees' suggestions and readers' comments.

This revision addresses comments and feedback made by the esteemed referees, as detailed by our responses below:

Responses to the Respected Referee 1	
Referee 1:	The manuscript "Generalized Hodgkin-Huxley Model Captures Human P2X and AMPA Receptor Currents" develops a modified Hodgkin and Huxley framework to model ionotropic receptors. A key benefit in the model design is 5 different sets of equations that are designed to capture receptor-specific cooperativity, binding kinetics, and desensitization. The model is then used to fit several P2X (7 subtypes) and mGluA1 receptors. In all cases, human receptor data was fit, thus increasing the importance of the results. The development of the model is excellent, as well as the description in the methods of each piece of all the equations with an intuitive description of how to interpret them. Once a receptor subtype is fit, the analysis of activation and inactivation kinetics (as well as other model equations) is accompanied by a very intuitive description of the function of the model. For example, in the results of Figure 3, "Such detailed mapping for rate constants across a wide range of ATP concentrations lays a foundation for understanding how each receptor plays its role in physiological conditions and what happens when the receptor is dysregulated in pathological states". I could not agree more! The formalism of this new model will surely lead to novel discoveries in the detailed biophysics of pathological states and with this has the potential to lead to targeted drug discoveries. To summarize, the new and novel model is sure to impact research on ionotropic receptors in health and disease, will provide excellent insight into physiological mechanisms, and is highly original. I do have one major concern on data sharing, otherwise, I only have minor concerns below: Major: 1) One major concern is with providing enough information and sharing enough

	data that others could extend this work. This would surely benefit the scientific community and be more in line with current data sharing requirements in both Europe and at the NIH. Major: 1.1) Can the authors provide the source code for the implementation of the model equations? Likewise, can the authors provide the source code used to fit the model? In the former case, the implementation of the model is not mentioned aside from equations, and in the latter case it is briefly outlined as being done in Matlab. This could all be shared in, for example, a GitHub repository.
Our Reply:	We really thank the respected referee for their careful reading of our manuscript and their valuable comments. We also thank the reviewer for highlighting the importance of resource sharing for reproducibility and extension. In response, we have created a publicly accessible GitHub repository containing all source code related to the model equations, parameter fitting routines, and simulation workflows, as used in this study. The repository includes all MATLAB scripts, detailed documentation, and a README describing the file structure and usage. The repository is available at: https://github.com/poshtkohi/gHH To address this point, we added the following sentences to the revised manuscript: In lines 283 to 284: “(the model code is publicly accessible as given in Supporting Information S3)” In lines 900 to 902: “S3 Code. All MATLAB source code of the gHH model, including parameter fitting and simulations, can be found on the GitHub page at https://github.com/poshtkohi/gHH. It comes with a README file that explains the source code hierarchy.”
Referee 1:	Major: 1.2) Throughout the manuscript, the model is fit to experimental data taken from a range of experimental manuscripts (8 or so, one for each of P2X and AMPA). What is not clear is how the authors extracted the raw current traces from those manuscripts (to perform the fit). Were the raw traces provided by the source experimental manuscripts as raw data files? Or did the authors somehow extract the raw data (current) from the published figures? It would be good to see a summary of how this was done (even if just in the supplemental). Finally, it would be good to provide raw data files with the currents extracted from the source manuscripts that were then fit.
Our Reply:	We appreciate this valuable suggestion and now clarify our procedure in the revised manuscript and supplementary files. The experimental current traces used for model fitting were digitised from published figures in the primary literature using WebPlotDigitizer, as raw data were not available from the original authors. Each dataset corresponds to a specific receptor subtype and agonist concentration, as documented in the main manuscript. All digitised datasets are now provided in the GitHub repository under the hP2XR-hGluAR-model/data folder (https://github.com/poshtkohi/gHH/tree/main/hP2XR-hGluAR-

	model/data). Each file is accompanied by a comprehensive README.md that details the contents, units (time in s or ms, current in nA, μA or pA as appropriate), and correspondence to experimental conditions. The following sentences were added to the revised manuscript: In lines 289 to 292: “Additionally, the experimental current traces were extracted by using WebPlotDigitizer from published figures. To access original data files, see Supporting Information S2, which includes all digested datasets with units and experimental conditions.” In lines 896 to 899: “S2 Data. All digitised datasets extracted from human experiments referenced in this article are fully described and documented at https://github.com/poshtkohi/gHH/tree/main/hP2XR-hGluAR-model/data. The public repository includes a README file that specifies the units and receptor subtypes with the relevant experimental conditions.”
Referee 1:	Major: 1.3) The supplemental data files need to be better described and organized. I opened a few of them and they seem to be text (.dat) files with two columns of numbers. It would be nice to know, (i) what does each file correspond to, there are many .dat files and I do not know what they refer to, and (ii) what are the units on the corresponding two columns of numbers in each .dat file? Without providing a better description of the organization and units in all the .dat files, they are not that useful.
Our Reply:	We agree and have significantly improved the description and organisation of all supplemental data files. The hP2XR-hGluAR-model/data folder in the GitHub repository (https://github.com/poshtkohi/gHH/tree/main/hP2XR-hGluAR-model/data) now includes a comprehensive README.md file, which lists:  • Each data file, its corresponding receptor subtype, agonist (ATP or Glu) concentration, and the figure/panel from which it was digitised • The precise units for every column (time in s or ms; current in nA, μA or pA) • Additional metadata to ensure that each file can be readily understood and reused This detailed documentation ensures all datasets are fully interpretable and reusable by the community. We added the following paragraph to the revised manuscript: In lines 896 to 899: “S2 Data. All digitised datasets extracted from human experiments referenced in this article are fully described and documented at https://github.com/poshtkohi/gHH/tree/main/hP2XR-hGluAR-model/data. The public repository includes a README file that specifies the units and receptor subtypes with the relevant experimental conditions.”
Referee 1:	Minor Concerns

	Minor: 1) There is a huge amount of model fitting (which in itself is great) and each fit is accompanied by an excellent description which gives both modelers and importantly non-modeler an intuitive understanding of how the model is operating. Yet, this is all buried in the results (which is fine). It might be nice to have a table summarizing the intuitive descriptions for how the model is working for all the P2X and AMPA fits? Maybe a table highlighting key points for each receptor, or key differences in the model between different types of receptors.
Our Reply:	We really thank the referee for bringing this point to our attention, which improves the readability of our manuscript. We agree that a concise overview will assist readers—particularly non-modellers—in comparing receptor phenotypes at a glance. We have therefore added a new subsection “Model summary and cross-receptor comparison” (Results section) and Table 2. The table lists, for each hP2X subtype and hGluA1 receptor:  (i) the gHH current-form equation that provided the best fit, (ii) qualitative activation speed, (iii) desensitisation profile, (iv) recovery time-scale, and (v) Distinctive kinetic features captured by the model. This addition distils the intuitive explanations given in the individual receptor sections into a single, reader-friendly reference. We hope the reviewer finds this helpful. The added section, in lines 751 through 759, can be seen as follows:

Table 2: Summary of the gHH model fitted to hP2X₁₋₇ and hGluA1 receptors.

Receptor	Best Current Form	Activation behaviour	Desensitisation / plateau	Recovery timescale	Distinctive kinetic feature
hP2X ₁	Eq. 7: Multiplicative activation, additive inactivation	Fast; peaks within < 50 ms at $\geq 4 \mu\text{M}$ ATP	Partial; shallow plateau at high ATP	Seconds	Rapid activation yet sustained residual current at low ATP
hP2X ₂	Eq. 9: Dual-pathway additive-multiplicative	Rapid and sustained activation; biphasic response depending on ATP	Partial inactivation with pronounced dip and recovery at varying ATP; plateau at sustained ATP	Seconds	Dual-pathway gating explains transient “sag-recovery” profile
hP2X ₃	Eq. 7: Multiplicative activation, additive inactivation	Ultra-fast (≤ 5 ms)	Near-complete within < 100 ms	Minutes; ATP-dependent (Fig. 13)	Long use-dependent desensitisation typical of nociceptors.
hP2X ₄	Eq. 8: Additive activation/inactivation	Moderate (1s-1.5s)	Tri-phasic; shallow mid-range plateau	Seconds	Bile-acid-sensitive triphasic inactivation reproduced.
hP2X ₅	Eq. 6: Additive activation, multiplicative inactivation	Fast at high ATP; slower at high ATP	Partial; ATP-dependent depth	Seconds	Strong cooperative exponents ($n_{1,2,3} \approx 2-3$) capture subunit synergy.
hP2X ₆	Eq. 6: Additive activation, multiplicative inactivation	Slow onset (≥ 200 ms)	Very slow decay	Seconds	Receptor remains open substantially longer, suggesting prolonged signalling roles.
hP2X ₇	Eq. 6: Additive activation, multiplicative inactivation	Slow (400ms, due to Unique delay in m_2 gate) then sustained.	Minimal; quasi-non-desensitising	Seconds	Plateau current without rundown reproduced over 28 s pulse. Irregular, highly ATP-dependent kinetics.
hGluA1	Eq. 10: Fully multiplicative gating	Ultra-fast (≤ 1 ms)	Rapid desensitisation (<5 ms)	Millisecond s	Exponential $\phi(A)$ term captures sub-millisecond recovery.

Referee 1:

Minor: 2) In future work it would be interesting to see a model run where ATP concentration is modulated in real time. The current model fits parameters and then calculates the current for one concentration of ATP. It would be great to see a model run for dynamic levels of ATP as can occur during action potential firing (in neurons and

	cardiac myocytes for example).
Our Reply:	We agree that applying time-varying agonist profiles represents an important and physiologically relevant extension of our framework. To address this, we have added the following statement to the Discussion (in lines 852 to 857) highlighting the model's capacity to simulate receptor currents under dynamic ATP or glutamate waveforms: "The gHH model can simulate scenarios under which either ATP or glutamate varies in time, thanks to the explicit dependence of its rate constants on agonist concentration. Therefore, our framework enables building complex models that simulate the interaction of ligand-gated receptors under dynamic ATP levels with voltage-gated ion channels during repetitive firing of action potentials in synaptic transmission through neurons and the cardiac cycle through cardiac myocytes [90,91]."
Referee 1:	Minor: 3) Fig 1b. What are the solid and dotted lines in Fig1b? Do they somehow encapsulate "The model allows for precisely replicating a wide spectrum of receptor dynamics consisting of activation, inactivation, desensitization and recovery in different physiological condition".
Our Reply:	We thank the referee for pointing out the need for clarification. We have updated the caption of Fig. 1b to explicitly state that the dotted lines show which gates the current depends on (in lines 113-114) as follows: "Note that the dotted lines show which gates the current depends on."
Referee 1:	Minor: 4) In Table 1, how were hGluA1 parameters estimated? In the results, the authors are fitting data from a manuscript on hP2X subtypes? Does this this data also contain hGluA1 currents?
Our Reply:	We thank the referee for raising this point. Table 1 summarises fitted parameters for both hP2X and hGluA1 receptors, each based on experimental data as detailed in the corresponding subsections of the Results. For the hGluA1 model, we used data reported by Coombs et al. (2012), which presents experimental recordings for human GluA1 receptors at three different glutamate concentrations. This dataset is entirely distinct from the P2X receptor data. As indicated in the Results section ("Human GluA1 model"), the optimal model form was selected and parameters were fitted specifically to the experimental data by Coombs et al. (2012). Reference used in this reply: Coombs ID, et al. Cornichons modify channel properties of recombinant and glial AMPA receptors. J Neurosci. 2012;32:9796-9804.
Referee 1:	Minor: 5) In general, the authors did an excellent job of giving a narrative explanation of how the equations work. One minor question, in the first order equation 1-4, what are the units on agonist concentration "A"? Does it matter, I am not sure? Is it molar?
Our Reply:	We thank the referee for this question. The units for the agonist concentration "A" (as well as rate constants and time) are specified separately for each receptor model in the Results section (inside figures for rate constants and currents) and detailed further in Supporting Information S1. For clarity, we have added the following statement to in lines 135-136: "Note that the units for agonist (A), rate constants and time are given separately for each receptor model in the results section and Supporting Information S1."

	This ensures that the units are explicit for all models discussed.
Referee 1:	Minor: 6) Fig5. The figure is good as it is but want to suggest merging m1/m2 into one plot and h1/h2 into another? That could be done for all m1/m2 plots throughout the manuscript? It is up to the authors.
Our Reply:	We thank the reviewer for this suggestion. We chose to display m_1 and h_1 (as well as m_2 and h_2) in vertically aligned pairs, rather than merging all activation or inactivation gates into single plots, to help readers more easily relate each activation gate with its corresponding inactivation gate. This layout was intended to facilitate smoother interpretation alongside the textual explanations in the manuscript. We hope the reviewer agrees that this approach maintains clarity for both specialist and non-specialist readers.
Referee 1:	Minor: 7) Fig13 shows recovery time for hP2X ₃ receptor. Is this recovery unique to hP2X ₃ versus 1/2? This figure shows recovery to a brief pulse of ATP (2 s), what about a longer application? Does that alter recovery?
Our Reply:	We thank the reviewer for this question. The prolonged recovery is unique to hP2X ₃ . When we applied the same recovery protocol to hP2X ₁ and hP2X ₂ models, both channels returned to baseline within seconds, not minutes. For this purpose, we have added Supporting Information S1.4. Additionally, repeating the simulation with a 30-s pulse produced virtually the same recovery at 1 μ M ATP for a 2-s pulse for hP2X ₃ R. Because hP2X ₃ desensitises very fast, extending the pulse does not drive it further into the inactivated state.
Referee 1:	Minor: 8) The sensitivity analysis to the P2X models is an excellent plus. Yet, a 0.1 or 1 percent tweak in the parameters seem small but this may be acceptable given the particular sensitivity analysis performed. If this is the case, it should be mentioned.
Our Reply:	We thank the reviewer for this helpful observation. In response, we have increased the parameter perturbation in our sensitivity analysis from 1% to 10% in both the revised main manuscript (see Fig. 31) and in all supplementary sensitivity analysis figures (Supplementary Information S1). As shown, even with a 10% change in gHH model parameters, the sensitivity analysis results remain similar, confirming that the optimiser has produced stable parameter estimates with low sensitivity to variation.
Referee 1:	Minor: 9) In Fig30, the authors should explain what "Relative Sensitivity" is. Is it the change in the parameter before/after changing model fits by 0.1 and 1 %? Basically, I am not sure what the Y-axis represents?
Our Reply:	We thank the reviewer for requesting clarification. The sensitivity analysis (SA) was performed by varying each rate constant individually (with all others held fixed). Then we measured the resulting change in the simulated whole-cell current (for a specific level of ATP). Therefore, the Y-axis in the SA graphs shows the proportional change in the whole-cell current when a single parameter is perturbed. Revisions made: In the revised manuscript, we have expanded this explanation in the "Sensitivity analysis of the model" section to state that 1. Lines 728 to 730 as follows: "we carried out sensitivity analysis (SA) by varying each rate constant individually (with all others held fixed). Then we measured the resulting change

	in the simulated whole-cell current (for a specific level of ATP).” 2. Lines 737 to 746 as follows: “Note that the Y-axis in Fig. 31 the SA graph shows the proportional change in the whole-cell current when a single parameter is perturbed. Our aim here was simply to demonstrate that the fitted model remains stable when rate constants deviate from their optimal values by amounts comparable to experimental uncertainty. We therefore adopted the classical “one-parameter-at-a-time” (OPAAT) approach: each rate constant was perturbed by 0.1 % to obtain a numerically stable local derivative, and by 10 % to bracket the ≈5–15 % error typical of ligand-gated channel fits. The Y-axis in Fig. 31 reports the relative sensitivity ($\Delta I/I$), so all curves are directly comparable. The near-overlap of the 0.1 % and 10 % traces confirms that the model output is governed by first-order behaviour and is therefore robust to moderate parameter variation.”
--	---

<h2>Response to the Respected Referee 2</h2>	
Referee 2:	This study proposes a generalized version of the Hodgkin-Huxley model that incorporates ligand-gated receptors, rather than being limited to voltage-gated channels. It aims to provide a unified, mechanistic mathematical framework for understanding the gating properties of P2X receptors and glutamate receptors. The model extends the traditional Hodgkin-Huxley formulation by including two activation gates, two inactivation gates, and coefficients that capture cooperative ligand binding. This novel approach is anticipated to be broadly applicable to physiological research and drug discovery. The authors present a thorough evaluation of model fits for several ATP-dependent P2X channel isoforms across varying ATP concentrations, as well as for a glutamate receptor channel. While the results highlight the versatility of the proposed modeling framework, several important considerations merit discussion. These include the model evaluation strategy and the tradeoffs involved in using this approach compared to established Markovian models. Additionally, there is some concern that the fitted models, in their current form, may not yield sufficiently novel insights to engage a broad Journal of Physiology readership. Major: A robust discussion of advantages and disadvantages of existing Markov models, like from Mackay et al, would be helpful in placing the novelty of this method in context (https://journals.plos.org/ploscompbiol/article?id=10.1371/journal.pcbi.1005643) Minor: Line 715 argues that their unified framework could obviate the fitting of 'complex, multi-state Markov models'. In my opinion, it would probably be helpful to discuss the advantages and limitations of both approaches. As I see it, the model proposed here could be more phenomenological in nature, but cheaper to parameterize, whereas a Markovian model may be more mechanistic, albeit have more free parameters to determine.

Our Reply:	Our response to the major concern We thank the reviewer for requesting a clearer comparison with existing multi-state Markov models. Accordingly, we have inserted a new, data-driven discussion block in the Discussion section (in lines 770-813 of the revised manuscript). The paragraph (i) introduces the “state-creep” problem with concrete numbers, using the P2X₄ analysis of Mackay et al.—22 states/58 parameters expanding to 34 states/65 parameters—to illustrate how complexity can escalate, and then provides parallel examples for P2X₇ and NaV1.5 channels; (ii) summarises three systematic drawbacks of such large schemes—parameter non-identifiability (supported by Bayesian MCMC studies), computational stiffness, and subjective topology choice (speed/identifiability trade-off analysed by Fink & Noble – please see the revised text for this reference); and (iii) explains how our four-gate generalised Hodgkin–Huxley (gHH) framework overcomes those issues while reproducing the same macroscopic kinetics gHH-model-paper-revised1. We believe this extended discussion now places the novelty and scope of the gHH model in a balanced historical context, exactly as requested. Response to the minor concern The revised paragraph also now provides an explicit, even-handed comparison of the two approaches (lines 808–813). We state that finely grained Markov schemes “remain invaluable when single-channel or high-resolution structural data justify explicit intermediates such as ligand-binding order, pore-dilation steps, or mutation-specific pathways,” but that the gHH formulation offers “a complementary, computationally lean description that preserves mechanistic insight while making large-scale simulations and drug-discovery workflows tractable”. These added sentences acknowledge the mechanistic depth of Markov models while highlighting the parameter-economy, identifiability and speed advantages of the gHH framework, thereby addressing the reviewer’s request for an exposition of both the strengths and limitations of each modelling strategy. The exact added text comes below (in lines 770-813): “Many recent studies illustrate a state-creep phenomenon in Markov-type ion-channel modelling: to capture ever-finer kinetic subtleties, authors successively append layers of closed, open, desensitised or dilated states until the scheme becomes unwieldy. For the P2X₄ receptor, Mackay et al. first tried a 22-state network with 58 adjustable parameters; only after expanding to a 34-state, 65-parameter diagram were fast activation and deactivation successfully reproduced [7]. A comparable escalation is seen in P2X₇ work, where an eight-state naïve/sensitised model grew to twelve states to explain biphasic currents and pore dilation [5,10]. Voltage-gated channels follow the same trend: contemporary Na_v1.5 descriptions employ eight to thirteen states [16]. Large state spaces pose three recurrent problems, including, identifiability, computational burden, and topology choice. Bayesian MCMC analyses have shown that, for large Markov schemes, many different rate sets reproduce the same macroscopic data, leading to practical non-identifiability and divergent mechanistic interpretations [19,73,74]. Stiff ODE systems with dozens of coupled equations slow whole-cell or tissue simulations to the point where uncertainty analysis becomes prohibitive [74]. Because most receptors lack complete structural or single-channel constraints, deciding which transitions to include is necessarily subjective, and alternative diagrams often fit equally well. Designing a Markov model is itself burdensome: investigators must first hypothesise a state diagram consistent with ligand stoichiometry, then wade through a combinatorial swarm of alternative topologies. Even with automated searches, iterating among candidate networks and constraints remains a labour-intensive task that demands equal parts biophysical insight and kinetic-modelling expertise. The generalised Hodgkin–Huxley
--

	framework presented herein addresses these issues directly. With only four dynamic variables and a handful of integer exponents it reproduces the kinetics of seven hP2X subtypes and hGluA1 AMPA receptors—behaviours that previously demanded 8- to 34-state schemes. All parameters map onto biologically meaningful quantities (activation, inactivation and recovery rates plus cooperativity indices), which greatly clarifies sensitivity and mutation analyses. Importantly, because the number of parameters is limited, these parameters are identifiable with routine data fitting, and their values can be robustly estimated from experiments. Reducing the model to a few state variables confers substantial computational benefits. Simulations can run sufficiently quickly that Monte-Carlo uncertainty propagation and high-throughput virtual-ligand screening become practicable on standard research hardware, as emphasised in the systematic assessment of speed versus complexity by Fink & Noble [74]. At the same time, identifiability improves because the compact parameter set can be constrained by conventional voltage- or ligand-jump protocols rather than exhaustive factorial designs. Versatility is retained through interchangeable current expressions that invoke multiplicative, additive or hybrid gating logic. These variants emulate with excellent accuracy properties like cooperativity, independent branch gating, use-dependence, and bi-exponential or sigmoidal responses without redrawing network diagrams. Finely grained Markov models remain invaluable when single-channel or high-resolution structural data justify explicit intermediates such as ligand-binding order, pore-dilation steps, or mutation-specific pathways, but the gHH framework offers a complementary, computationally lean description that preserves mechanistic insight while making large-scale simulations and drug-discovery workflows tractable.”
Referee 2:	Major: Hodgkin-Huxley has been a useful phenomenological model for describing ion channel activity, and as the author indicates, its simplicity limits its interpretability. The authors' extended model adds gating terms, but still seems to be lacking in physical factors such as reversibility and detailed balance (such as in Figure 1B) that arises naturally in Markovian models. Would these omissions limit the interpretability of this extended model?
Our Reply:	We thank the reviewer for emphasising that explicit microscopic reversibility and detailed balance can be important for certain mechanistic questions. Interpretability, however, lies on a continuum: classic Hodgkin–Huxley models may be too simple, whereas large multi-state Markov (MM) schemes can become so elaborate that they obscure intuition. Our work deliberately explores the middle ground—a parsimonious four-gate generalised HH scaffold (gHH) that preserves whole-cell accuracy while keeping all parameters tightly identifiable. The five points below explain why enforcing full detailed balance is unnecessary for the present macroscopic study and summarise the tests we performed to verify that conclusion.  1. Scope of the present study. We set out to test whether this minimal gHH framework can reproduce macroscopic P2X and AMPA currents across an $\sim 10^3$-fold agonist range with only 8 kinetic parameters—far fewer than the 20–40 typically required by MM schemes. Extracting microscopic free energies—where strict detailed balance is indispensable—was beyond the present scope. 2. Introducing an explicit backward dependence. To probe the reviewer’s concern, we implemented a pseudo-reversible variant in which each backward activation rate is multiplied by its cognate inactivation gate (Eqs S1.1–S1.2). This adds a bidirectional term at the ODE level, increasing physical symmetry without enforcing the full loop products demanded by detailed balance. 3. Empirical test on three receptors. We refitted the variant to representative data

sets for hP2X₄, hP2X₇ and hGluA1 (Table S1.16; Figs S1.8 – S1.10). In every case the sum-of-squared-error increased (Δ SSE +1 % to +49 %) even though both models contain the same number of free parameters. Standard information criteria therefore favour the simpler forward-dominant formulation.

4. Interpretability is not compromised at the macroscopic level. The forward-dominant gHH already captures activation, desensitisation and recovery with tight confidence intervals. Adding reciprocal terms only inflates parameter uncertainty and, paradoxically, reduces interpretative clarity—illustrating that excess mechanistic detail can hinder, rather than help, macroscopic understanding.
5. Flexibility for future mechanistic work. The pseudo-reversible equations are provided in Supplementary Section S1.2; activating them is a one-line code change. Researchers requiring full thermodynamic loop closure can therefore adopt that option without altering the wider framework.

Revisions made:

- *Discussion* lines 819-826 now clarify the optional feedback as follows:

“Even though the gHH equations are written in a forward-dominant form, they can be augmented with reciprocal (h-dependent) backward fluxes (Eqs S1.1–S1.2). This feedback adds an explicit return path between activation and inactivation gates—approximating microscopic reversibility—yet it does not enforce the stricter loop constraints of detailed balance. Implementing the variant for hP2X₄, hP2X₇ and hGluA1 increased the fitting error in every case, indicating that the simpler formulation already captures the macroscopic kinetics; the feedback option remains available for future studies that require a closer link to thermodynamic cycles.”

- *Discussion* lines 814-817 now better clarify the position of our framework among HH and Markov formalisms as follows:

“Although classical Hodgkin–Huxley schemes are often called phenomenological—they reproduce observed currents without enumerating every microscopic state—their α/β rate functions still correspond to physically meaningful transitions; our four-gate generalisation preserves that semi-mechanistic link while remaining far more compact than large Markov models.”

- Supplement S1. 2 contains the equations, statistical comparison and representative traces as follows:

The forward-dominant gHH model treats the α and β rate pairs phenomenologically; each gate has a backward flux, but that flux is independent of the companion inactivation variable. To create an explicit reciprocal coupling—without introducing extra states—we multiply every backward rate of the activation gates by its cognate inactivation variable, yielding Eqs S1.1–S1.2. This modification restores a bidirectional link at the ODE level but does not by itself impose strict detailed balance (the product constraints on multi-gate cycles). We therefore refer to it as the *h-feedback* or *pseudo-reversible* variant. Fits and statistics for the two formulations are compared below (Table S1.16, Figs. S1.8–S1.10).

$$\frac{dm_1}{dt} = \alpha_{m_1}(A) \times A \times (1 - m_1) - \beta_{m_1}(A) \times [1 + \varphi(A)] \times m_1 \times h_1 \quad (\text{Eq. S1.1})$$

$$\frac{dm_2}{dt} = \alpha_{m_2}(A) \times m_1^{n_1} \times (1 - m_2) - \beta_{m_2}(A) \times [1 + \varphi(A)] \times m_2 \times h_2 \quad (\text{Eq. S1.2})$$

Table S1. 16: Comparison of the sum-of-squared-errors (SSE) obtained with the forward-only gHH model versus the reversible h-feedback variant (Eqs S1.1–S1.2). Δ SSE is expressed relative to the forward-only value; positive numbers indicate a worse fit. The h-feedback modification therefore offers no statistical advantage for any of the three receptors.

Receptor	Agonist	SSE forward-only	SSE + h-feedback	Δ SSE	Preferred
hP2X ₄	ATP 50 μ M	0.7133	0.9332	+24%	forward
hP2X ₇	ATP 3 mM	5.412×10^{-4}	8.039×10^{-4}	+49%	forward
hGluA1	Glu 10 mM	4.6334	4.6820	+1%	forward

Please refer to the revised Supporting Information (Section S1. 2) for Figs. S1.8–S1.10, which are located on pages 8–10.

In summary, while detailed balance is important for some single-channel questions, our results show it is not required to interpret or predict ligand-gated whole-cell currents—indeed, enforcing it here would sacrifice both parsimony and statistical performance.

Referee 2:

Minor: Line 691 Could the sensitivity analyses be explained in greater detail? In this elaboration, discussion justifying the advantage of their approach relative to more conventional ones (e.g. a figure of merit is evaluated across a parameter range) would be appreciated. The sensitivity analyses are based on comparisons of predicted currents for perturbed parameters relative to those predicted with best-fit parameters. The parameter perturbations (0.1 and 1% respectively) seem very small, relative to the experimental error one might encounter with determining those parameters.

Our Reply:

We appreciate the referee for this point. We have expanded the sensitivity-analysis subsection to clarify purpose and methodology:

- **Objective.** Our goal was to test robustness, not to perform an exhaustive global scan.
- **Method.** Using a one-parameter-at-a-time (OPAAT) protocol, we varied each rate constant by 0.1% (for a precise local derivative) and 10% (to match realistic experimental error).
- **Metric.** We report the relative sensitivity ($\Delta I/I$), which normalises current magnitude and makes parameters directly comparable.
- **Result.** Figure 31 now shows both perturbation levels; the curves nearly coincide, demonstrating that the model's behaviour is first-order in this neighbourhood and therefore robust.

This focused test fulfils its intended purpose—showing that moderate parameter deviations do not compromise model predictions—without incurring the computational

	cost of a full multidimensional grid search. We believe this added explanation addresses the referee's request. We also increased the 1% perturbation to 10%. Revisions made: In the revised manuscript, we have expanded this explanation in the "Sensitivity analysis of the model" section to state that  1. Lines: 728 to 730 as follows: "we carried out sensitivity analysis (SA) by varying each rate constant individually (with all others held fixed). Then we measured the resulting change in the simulated whole-cell current (for a specific level of ATP)." 2. Lines: 737 to 746 as follows: "Note that the Y-axis in Fig. 31 the SA graph shows the proportional change in the whole-cell current when a single parameter is perturbed. Our aim here was simply to demonstrate that the fitted model remains stable when rate constants deviate from their optimal values by amounts comparable to experimental uncertainty. We therefore adopted the classical "one-parameter-at-a-time" (OPAAT) approach: each rate constant was perturbed by 0.1 % to obtain a numerically stable local derivative, and by 10 % to bracket the ≈5–15 % error typical of ligand-gated channel fits. The Y-axis in Fig. 31 reports the relative sensitivity ($\Delta I/I$), so all curves are directly comparable. The near-overlap of the 0.1 % and 10 % traces confirms that the model output is governed by first-order behaviour and is therefore robust to moderate parameter variation."
Referee 2:	Minor: Line 254 indicates that only one current form is 'appropriate' for each channel. How was this determined exactly? Also, do they try models with fewer or more gates to assess the quality of fitting?
Our Reply:	For each receptor we fitted all five candidate current equations (Eqs. 6–10) to the entire set of current traces and selected the form with the lowest summed-squared error (SSE). In every case the chosen form uniquely reproduced peak, desensitisation, and recovery phases, which is now stated in the main text, in lines 272-274 as follows: "For each receptor we fitted all five candidate current forms (Eqs. 6–10) to the full multi-trace data set and retained the form with the lowest SSE; only one form provided a satisfactory fit across all agonist concentrations for that receptor." We also created a reduced model containing only one activation (m_1) and one inactivation (h_1) gate, refitted it, and found that it failed to capture either the rising phase or the late recovery for all receptors tested (please see Supporting Information S1.3, Fig. S1.11). Conversely, adding additional gates did not improve fit quality but introduced parameter non-identifiability. These results demonstrate that the four-gate configuration is the minimal—and sufficient—structure for satisfactory fits across all 8 receptors and 38 current recordings. Corresponding statements have been added to the manuscript in lines 279–282 as follows: "We also confirmed that removing activation gate (m_2) and inactivation gate (h_2) degraded the fit markedly (see Supporting Information S1.3, Fig. S1.11), indicating that the four-gate architecture is the minimal structure that reproduces all 38 human P2X_{1,7} and GluA1 recordings."

Referee 2:	Minor: Line 311 indicates that optimal beta_m2 values vary non-monotonically with ATP concentration. Could this be evidence of there being different states, such as low affinity and high affinity states that wouldn't necessary be captured by assuming one state for m2? In Line 397, this is alluded to in the suggestion that there may be multiple energetic barriers.												
Our Reply:	Thank you for raising the point about the non-monotonic profile of β_{m_2}. The peak in β_{m_2} near 1 μM ATP suggests more than a single, uniform m_2 de-activation route. This likely reflects ATP stabilising m_2 in different affinity states:    ATP regime Putative sub-state Effect on β_{m_2}     Low ($\leq 0.3 \mu\text{M}$) High-affinity, partial binding β_{m_2} low (slow closure)   Intermediate ($\sim 1 \mu\text{M}$) Mixed occupancy (1–2 sites) β_{m_2} peaks   High ($\gg 1 \mu\text{M}$) Fully liganded, lower affinity β_{m_2} decreases    Such affinity switching is supported by structural and functional studies (Mansoor et al., 2016; Wang et al., 2018), which show distinct ATP-bound states and conformational changes affecting channel kinetics. Our single m_2 gate serves as an effective parameter capturing these underlying mechanisms, consistent with the interpretation in Line 397 of the original manuscript. We added a statement to the revised text in lines 338-340 as follows: “The peak in β_{m_2} at 1 μM ATP likely marks a shift between high- and low-affinity de-activation: partial occupancy favours closure, while both low and high ATP stabilise alternative conformations, lowering β_{m_2} [39,40].” We also made changes manuscript text in lines 425-428 as follows: “These biphasic changes which are highly ATP-dependent suggest multiple energetic barriers (such as multiple conformational or binding steps), affinity states or competing pathways that finely tune the hP2X₂R gating properties, which our gHH model captures simply and universally.” References used for this reply: Mansoor SE, Lü W, Oosterheert W, et al. X-ray structures define human P2X₃ receptor gating cycle and antagonist action. Nature. 2016;538:66-71. doi:10.1038/nature19367 Wang J, Wang Y, Cui W-W, et al. Druggable negative allosteric site of P2X₃ receptors. Proc Natl Acad Sci USA. 2018;115:4939-4944. doi:10.1073/pnas.1800907115	ATP regime	Putative sub-state	Effect on β_{m_2}	Low ($\leq 0.3 \mu\text{M}$)	High-affinity, partial binding	β_{m_2} low (slow closure)	Intermediate ($\sim 1 \mu\text{M}$)	Mixed occupancy (1–2 sites)	β_{m_2} peaks	High ($\gg 1 \mu\text{M}$)	Fully liganded, lower affinity	β_{m_2} decreases
ATP regime	Putative sub-state	Effect on β_{m_2}											
Low ($\leq 0.3 \mu\text{M}$)	High-affinity, partial binding	β_{m_2} low (slow closure)											
Intermediate ($\sim 1 \mu\text{M}$)	Mixed occupancy (1–2 sites)	β_{m_2} peaks											
High ($\gg 1 \mu\text{M}$)	Fully liganded, lower affinity	β_{m_2} decreases											
Referee 2:	Minor: Line 133. are there any constraints used to determine kinetic parameters when the system is in steady state? In other words, are any measures taken to hold some states fixed, such that a smaller number of parameters can be optimized for a given observable?												

As it stands, Eqn 9 has at least four free parameters to fit to a 1D line, therefore it is very likely that the parameter solutions are not unique.

Our Reply:

We thank the referee for this question. Steady-state constraints were not imposed because, for ligand-gated receptors, the agonist $A(t)$ is an external, rapidly time-varying drive: each protocol jumps from 0 to a finite value and back again, so the gates spend virtually the entire experiment in transient, not equilibrium, conditions. This contrasts with classic voltage-clamp Hodgkin-Huxley analysis of voltage-gated channels, where the membrane potential is held long enough for true steady state and α/β ratios can be fixed analytically. Instead, we identify all rate constants and exponents by fitting the complete current waveform at multiple agonist concentrations simultaneously, with every parameter bounded to physiologically plausible ranges. Because the two multiplicative terms in Eq. 9 shape different temporal phases of the response (activation rise, peak amplitude, desensitisation, and recovery), the multi-trace fit provides sufficient independent information for a unique and reproducible solution without needing equilibrium constraints.

To test the referee's suggestion, we clamped the two m_2 and h_2 gates at steady state (namely, $\frac{dm_2}{dt} = 0$ and $\frac{dh_2}{dt} = 0$), solved Eqs 3-4 for m_2 and h_2 in terms of m_1 , and m_1 and m_2 , respectively, and re-fitted the reduced system (plots below). The constrained version failed to reproduce either the rising phase or the late recovery at 100 μM and 300 μM ATP (dashed red vs. black). Thus, the extra degrees of freedom in Eq. 9—and the full, non-equilibrium formulation of Eqs. 3-4—are both necessary and uniquely identifiable under our multi-trace fitting procedure.

Referee 2: **Minor:** Line 133. How (or why) was the modulation factor of $p(A)=\exp(-A)$ chosen? It seems like the primary effect would be to maintain a small or potentially negative value for states like m_1 at low substrate concentrations.

Our Reply: We appreciate the referee for asking this question. We chose $\varphi(A) = e^{-A}$ for three reasons:

1. Physiological logic. When agonist is removed ($A \approx 0$), $\varphi = 1$ and the terms $\alpha_n \times [1 + \varphi]$ and $\beta_m \times [1 + \varphi]$ are doubled, allowing the model to reproduce the fast recovery of receptors such as hGluA1. At high agonist, $\varphi \rightarrow 0$, so the backward rates return to baseline, matching the experimentally observed slowing of recovery/desensitisation.
2. Mathematical parsimony. e^{-A} is the simplest single-parameter function that is monotonic, bounded between 0 and 1, and continuously differentiable over the full concentration range. It therefore adds the required non-linearity without introducing extra fitting parameters or equations.
3. Numeric stability. Because $\varphi(A)$ is always positive and ≤ 1 , the factor $[1+\varphi(A)]$ never drives gating variables negative; it only scales the rate constants, ensuring stable integration of the four-ODE system.

We have added a brief explanation of these points in the revised manuscript (lines 136-

	141) as follows: “We selected $\varphi(A) = e^{-A}$ because it is the simplest one-parameter, monotonic function that (i) equals 1 at agonist wash-out ($A \rightarrow 0$) and therefore boosts the recovery/de-inactivation (β_{m_i} and α_{h_i}) rates by a factor of two, and (ii) decays smoothly to 0 at high agonist, so the backward terms revert to their baseline forms. This dual behaviour lets the four-ODE gHH system reproduce the fast recovery of AMPA receptors after brief pulses without adding extra states or rate constants.”
Referee 2:	Minor: Line 134 Is there a physical basis for determining or interpreting the values of n_i? Clearly they add nonlinearity to the rate equations, akin to cooperativity, but how cooperativity arises from the gates is unclear. For instance, how does $n_4=5$ for hP2X₄ channels that are trimeric in nature?  • Line 282 - are the values of n constrained in the optimization • strategy?
Our Reply:	We thank the referee for making these questions. All exponents n_i were restricted to integer values 0–5 during optimisation; the algorithm therefore selects the smallest exponent that reproduces the data. The 0–5 window follows classical practice: Hodgkin & Huxley found a 4th power sufficient for g_K and noted that a 5th or 6th power would only sharpen the fit without changing interpretation (Hodgkin & Huxley, J. Physiol. 117, 500 (1952)). In our framework the n_i behave like Hill coefficients—they quantify the overall steepness (“effective cooperativity”) of the macroscopic gating process rather than the literal number of subunits. Thus a fitted $n_4 = 5$ for a trimeric hP2X₄ receptor simply reflects an especially steep activation–inactivation interplay that is still captured parsimoniously within the 0–5 grid. We added the following statement to lines 257-262 to make both the optimisation constraint and the interpretive rationale explicit: “Therefore, during optimisation each exponent n_i was constrained to the integer range 0–5. This bracket is wide enough to span plausible stoichiometries (0 = no coupling; 1 = independent; 2–3 \approx dimer/trimer; 4–5 = highly cooperative, akin to the classical 4th-power in the Hodgkin–Huxley K⁺ model [2]), yet narrow enough to prevent over-fitting; thus the $n_4 = 5$ found for hP2X₄ signals an especially steep effective cooperativity, not five physical subunits (see Table 1).”
Referee 2:	Minor: Line 318 However, ah_2 shows a district sensitivity <-- typo
Our Reply:	We thank the referee for pointing out this typo; it has now been corrected to “distinct sensitivity” in the revised text (line 345).
Referee 2:	Minor: Considerable heterogeneity in receptor expression is anticipated, how would the strategy reflect this in its parameter optimization?
Our Reply:	We thank the reviewer for highlighting expression heterogeneity. Our optimisation proceeds in two tiers: for every recording we first obtain an individual conductance scaling factor from the peak current produced by the highest agonist pulse and keep that value fixed for the other pulses from the same cell, so variations in receptor copy number are absorbed at this stage; the full data set is then used to refine a single set of α- and β-rate functions, yielding kinetic parameters that describe receptor gating independent of

	expression level. A clarifying sentence describing this procedure has been added to the Model-fitting section in lines 262-266 as follows: "Note that g_{hIR} is obtained from the maximal current at the highest ligand (ATP or Glu) pulse. Because this conductance is re-estimated for every individual cell (from its own peak current) and then held fixed for that cell's remaining ligand levels, the procedure implicitly captures cell-to-cell variability in receptor expression while still allowing the kinetic rate constants to be fitted globally."
Referee 2:	Minor: Ln 226 How is goodness of fit determined and used in this study? I believe this is implied in the Model fitting section (i.e. line 245).
Our Reply:	We appreciate the referee for asking this question. We now state this explicitly in the Model-fitting subsection in lines 250-252: "Once these phases get completed, the parameter set that best minimises the error—defined as the goodness-of-fit (sum of squared errors, SSE) between model simulations and experimental data—is selected." SSE is therefore the objective function throughout the optimisation, and the exponent set / current form with the lowest SSE is chosen.
Referee 2:	Minor: Ln 499 The authors indicate that the recovery time for hP2X3 was 'confirmed by human experimental studies'. Could this be briefly elaborated in the text, such as the cell type, whether they used primary cells, etc.
Our Reply:	We thank the referee for this point. We have added the requested information. The supporting data are from "Pratt, Emily B., et al. "Use-dependent inhibition of P2X3 receptors by nanomolar agonist." Journal of Neuroscience 25.32 (2005): 7359-7365.", who expressed human P2X₃ receptors in HEK293 cells (a standard immortalised cell line, not primary tissue) and measured recovery with whole-cell patch-clamp. After desensitisation with 30 μM ATP, full recovery occurred in approximately 8 minutes, matching our model prediction. Revisions made: We revised the manuscript text in lines 528-530 as follows: "This is confirmed by the work in [51], who recorded hP2X₃ receptors expressed in HEK293 cells and found that a saturating 30 μM ATP pulse required approximately 8 minutes for full recovery from desensitisation."
Referee 2:	Minor: Line 662 indicates that the functional form of $\phi(A)$ is exponential in A. Have the authors considered the possibility that A may also be time dependent, given that both nucleotides and glutamate can be degraded?
Our Reply:	We are thankful to the referee for point. Yes, in the gHH equations A is an input, so $\phi(A)$ is really $\phi(A(t))$. If one supplies a time-dependent agonist concentration that decays because of hydrolysis (ATP) or uptake (glutamate), the term $\phi(A(t))$ will reflect that decay automatically.

	In the present study we matched the experimental protocols, which use rapid solution exchange (for P2X) or very short synaptic-like pulses (for GluA1); within those time windows ligand degradation is negligible, and the perfusion system brings A down to baseline almost instantaneously. Therefore, we treated $A(t)$ as a square pulse. We have added a clarifying sentence to the manuscript in lines 697-699 (to make this explicit and to note that the framework is fully compatible with more elaborate ligand-time courses if required in future work) as follows: "In all simulations, A is a time-dependent input (namely $A(t)$) matching the experimental pulse. Any decay from degradation or uptake can be reflected in $A(t)$; here, brief ligand pulses are well approximated as square steps."
Referee 2:	Minor: Final paragraph beginning with Line 769 - the 'manual' strategy of determining the current form was not specified. I would advise at least discussing numerical strategies available to select between competing models. The line regarding supercomputers and parallel optimization is vague and extrapolative.
Our Reply:	We thank the referee for this helpful suggestion, which has led to a clearer and more precise description of our model selection strategy. The manuscript now states that we perform a manual search: each of the five current forms is fitted in turn, and the form with the lowest SSE is retained. We also note that the same step can be automated by systematically evaluating SSE across all candidate forms in a parallel optimisation framework (because the manual, sequential approach is time-consuming), removing the earlier vague reference to "supercomputers.", given the first author's published expertise in high-performance computing. See Discussion, revised in lines 884-888 as follows: "Concretely speaking, we fit each of the five candidate current forms to the data and keep the one that yields the lowest summed-squared error. This selection step can be automated by systematically evaluating SSE across all candidate forms in a parallel optimisation workflow to speed up determining the best mathematical structure for the receptor(s) in question."

Response to the Respected Editorial Team

Reviewing Editor/Senior Editor	Both referees agree that this work is likely to significantly impact the field. However, there are several issues that should be addressed. This includes issues regarding resource sharing as identified by referee 1 and further discussion on the approach used as identified by referee 2. Each referee also identified minor concerns that, if addressed, would improve the overall quality of the work.
Our Reply:	We thank the editorial team for highlighting the two overarching points raised by the referees. All requested revisions have now been completed:  • Resource sharing (Referee 1) – A public GitHub repository (link provided in S2 Data and S3 Code) now contains all MATLAB source files, digitised current traces, and detailed README documentation. Supplementary material has been

	reorganised and fully annotated to ensure complete reproducibility. • Approach comparison (Referee 2) – A new discussion block (lines 770-813) provides a balanced, data-driven appraisal of multi-state Markov models versus our four-gate generalised Hodgkin–Huxley framework, explicitly addressing versatility, identifiability, mechanistic detail and computational cost.• Reversibility and detailed balance (Referee 2): We now clarify in the Discussion (lines 814-826) and Supplementary Section S1.2 that the gHH model can optionally include backward rates to enforce reversibility and detailed balance, but statistical analysis shows this is not required to fit or interpret macroscopic P2X/AMPA currents, preserving both model parsimony and clarity. All additional minor comments have been answered point-by-point, with corresponding textual or figure changes noted in the manuscript. We hope these revisions meet the editorial expectations and respectfully submit the updated version for your consideration.
--	---

Dear Dr Poshtkahi,

Re: JP-RP-2025-288880R1 "**Generalised Hodgkin-Huxley Model Captures Human P2X and AMPA Receptor Currents**"
by Alireza Poshtkahi and Brian David Gulbransen

We are pleased to tell you that your paper has been accepted for publication in The Journal of Physiology.

Yours sincerely,

Kim Barrett
Senior Editor
The Journal of Physiology

IMPORTANT POINTS TO NOTE FOLLOWING ACCEPTANCE OF YOUR PAPER:

- You can help your research get the attention it deserves! Check out Wiley's free Promotion Guide for best-practice recommendations for promoting your work at: www.wileyauthors.com/eoo/guide. You can learn more about Wiley Editing Services which offers professional video, design, and writing services to create shareable video abstracts, infographics, conference posters, lay summaries, and research news stories for your research at: www.wileyauthors.com/eoo/promotion.

- If you would like to receive our 'Research Roundup', a monthly newsletter highlighting the cutting-edge research published in The Physiological Society's family of journals (The Journal of Physiology, Experimental Physiology, Physiological Reports, The Journal of Nutritional Physiology and The Journal of Precision Medicine: Health and Disease), please click this link, fill in your name and email address and select 'Research Roundup':
<https://www.physoc.org/journals-and-media/membernews>

EDITOR COMMENTS

Reviewing Editor:

Comments to the Author:

The authors have addressed the concerns raised by the previous referees. The study is expected to have a significant impact on the field.

REFeree COMMENTS

Referee #2:

I commend the authors on both their initial submission and their thorough and thoughtful revision. In this revised manuscript, the authors have addressed all of the two reviewers concerns.